# Horizon-free Reinforcement Learning in Adversarial Linear Mixture MDPs

**Kaixuan Ji**[1*]**, Qingyue Zhao**[2*]**, Jiafan He**[1]**, Weitong Zhang**[1]**, Quanquan Gu**[1]

[1]Department of Computer Science, University of California, Los Angeles
[2]Department of Computer Science and Technology, Tsinghua University
`kaixuanji@cs.ucla.edu,zhaoqy19@mails.tsinghua.edu.cn`
`{jiafanhe19,weightzero}@ucla.edu,qgu@cs.ucla.edu`

## Abstract

Recent studies have shown that the regret of reinforcement learning (RL) can be polylogarithmic in the planning horizon $H$. However, it remains an open question whether such a result holds for adversarial RL. In this paper, we answer this question affirmatively by proposing the first horizon-free policy search algorithm. To tackle the challenges caused by exploration and adversarially chosen reward over episodes, our algorithm employs (1) a *variance-uncertainty-aware* weighted least square estimator for the transition kernel; and (2) an *occupancy measure*-based technique for the online search of a *stochastic* policy. We show that our algorithm achieves an $\widetilde{O}\big((d + \log|\mathcal{S}|)\sqrt{K} + d^2\big)$ regret with full-information feedback[1], where $d$ is the dimension of a known feature mapping linearly parametrizing the unknown transition kernel of the MDP, $K$ is the number of episodes, $|\mathcal{S}|$ is the cardinality of the state space. We also provide hardness results to justify the near optimality of our algorithm and the inevitability of $\log|\mathcal{S}|$ in the regret bound.

## 1 Introduction

Learning in episodic Markov Decision Processes (MDPs) (Altman, 1999; Dann & Brunskill, 2015; Neu & Pike-Burke, 2020) is a key problem in reinforcement learning (RL) (Szepesvári, 2010; Sutton & Barto, 2018), where an agent sequentially interacts with an environment with a fixed horizon length $H$. Each action $a_t$ the agent takes at state $s_t$ incurs some reward $r(s_t, a_t)$, and takes it into the next state $s_{t+1}$. The agent will restart in the same environment after every $H$ time steps. Although the *curse of horizon* has been deemed as a challenge in episodic RL (Jiang & Agarwal, 2018), a recent line of works have developed near-optimal algorithms to achieve a regret with no polynomial dependence on $H$ for both tabular MDPs (Zhang et al., 2021a) and RL with linear function approximation (Zhang et al., 2021b; Kim et al., 2022; Zhou & Gu, 2022). This suggest that the regret of episodic RL can get rid of linear dependency on $H$. Nevertheless, these *horizon-free* algorithms are only applicable to learning MDPs where the reward function is either fixed or stochastic, yet in many real-world scenarios, we have cope with the adversarially changing reward (Even-Dar et al., 2009; Yu et al., 2009; Zimin & Neu, 2013). However, little is known about horizon-free RL in adversarial MDPs. Thus, the following question remains open.

*Can we design near-optimal horizon-free RL algorithms under adversarial reward and unknown transition with function approximation employed?*

In this paper, we affirmatively answer the question for linear mixture MDPs with adversarial reward under full-information feedback (Cai et al., 2020; He et al., 2022b). We propose a new algorithm termed **H**orizon-**F**ree **O**ccupancy-Measure Guided **O**ptimistic **P**olicy **S**earch (HF-O$^2$PS). Following Cai et al. (2020); He et al. (2022b), we use online mirror descent (OMD) to update the policies and value-targeted regression (VTR) (Jia et al., 2020; Ayoub et al., 2020) to learn the transition. Nevertheless, we show that the value-function-based mirror descent inevitably introduces the polynomial dependency on the planning horizon $H$ in the regret upper bound. To address this issue, inspired by Rosenberg & Mansour (2019a); Jin et al. (2020a) and Kalagarla et al. (2020), we use occupancy measure as a proxy of the policy and conduct OMD on the occupancy measures to update. Like Jin

---

*Equal Contribution

[1]Here $\widetilde{O}(\cdot)$ hides logarithmic factors of $H$, $K$ and $1/\delta$.

et al. (2020a), we maintain a confidence set of the transition kernel and utilize constrained OMD to handle the unknown transition. To achieve a horizon-free regret bound, we also adapt the high-order moment estimator in Zhou & Gu (2022) to stochastic policies and obtain a tighter Bernstein-type confidence set. The regret of our algorithm can be upper bounded by $\widetilde{O}(d\sqrt{K} + d^2)$ in the first $K$ episodes with high probability, where $d$ is the dimension of the feature mapping. To the best of our knowledge, our algorithm is the first algorithm to achieve horizon-free regret in learning adversarial linear mixture MDPs. Our three main contributions are summarized as follows:

- We propose a new algorithm, HF-O$^2$PS, for linear mixture MDPs with adversarial reward uniformly bounded by $1/H$. Compared to the previous works (e.g., Cai et al. (2020); He et al. (2022b)), HF-O$^2$PS use occupancy-measure-based OMD rather than direct policy optimization. HF-O$^2$PS also use the high-order moment estimator to further facilitate the learning of the transition kernel.

- Our analysis shows that HF-O$^2$PS achieves a regret bound $\widetilde{O}(d\sqrt{K} + d^2)$, where $K$ is the number of episodes and $d$ is the dimension of the feature mapping. As far as we know, HF-O$^2$PS is the first algorithm for adversarial RL achieving a horizon-free regret upper bound.

- We also provide hardness results in addition to the regret upper bound. First we show that an unbounded $\mathcal{S}$ will result in a lower bound asymptotically linear in $\sqrt{H}$, which justifies our assumption of $|\mathcal{S}| < \infty$. We also provide a minimax lower bound of $\widetilde{\Omega}(d\sqrt{K})$, which manifests the near optimality of HF-O$^2$PS.

**Notation** We denote scalars by lowercase letters, and denote vectors and matrices by lower and uppercase boldface letters respectively. We use $[n]$ to denote the set $\{1, \ldots, n\}$, and $\overline{[n]} = \{0, \ldots, n-1\}$. Given a $\mathbb{R}^{d \times d} \ni \boldsymbol{\Sigma} \succ \mathbf{0}$ and vector $\mathbf{x} \in \mathbb{R}^d$, we denote the vector's $L_2$-norm by $\|\mathbf{x}\|_2$ and define $\|\mathbf{x}\|_{\boldsymbol{\Sigma}} = \sqrt{\mathbf{x}^\top \boldsymbol{\Sigma} \mathbf{x}}$. We employ standard asymptotic notations including $O(\cdot), \Omega(\cdot), \Theta(\cdot)$, and further use $\widetilde{O}(\cdot)$ to hide polylogarithmic factors. Let $\mathbb{1}\{\cdot\}$ denote the indicator function, and $[x]_{[a,b]}$ denote the truncation function $x \cdot \mathbb{1}\{a \leq x \leq b\} + a \cdot \mathbb{1}\{x < a\} + b \cdot \mathbb{1}\{x > b\}$ where $a \leq b \in \mathbb{R}, x \in \mathbb{R}$. Let $\Delta(\cdot)$ represent the probability simplex over a finite set.

## 2 RELATED WORK

**Horizon-free RL** RL is once believed to be far more harder than contextual bandits. However, recent works has begun to overthrow this long-standing stereotype (Wang et al., 2020a). To achieve a fair comparison with CB, there are two options. One is to assume the total reward is bounded by one in each episode (Jiang & Agarwal, 2018). Under such assumption, It is possible to obtain entirely $H$-independent regret bounds in the tabular setting (Zhang et al., 2021a; 2022). Recent works further proposed horizon-free algorithms for linear mixture MDPs and linear MDPs (Zhang et al., 2021b; Kim et al., 2022; Chen et al., 2022; Zhou & Gu, 2022). Though near-optimal algorithms have been proposed to learn a Dirac policy with $\widetilde{O}(d\sqrt{K} + d^2)$ regret under linear function approximation (Zhou & Gu, 2022), and similar regret guarantees with no $\text{poly}(H)$ dependency has been established in various settings (Zhang et al., 2020; Tarbouriech et al., 2021; Zhou et al., 2022), any of the above work can not even learn a nontrivial categorical policy. Another assumption is to assume that the reward is uniformly bounded by $1/H$ (Assumption 2, Zhang et al., 2021a). We employ the later assumption to approach MDPs with large state space and adversarial reward and learn stochastic policies in a horizon-free manner.

**RL with adversarial reward** A long line of works (Even-Dar et al., 2009; Yu et al., 2009; Neu et al., 2010; 2012; Zimin & Neu, 2013; Dick et al., 2014; Rosenberg & Mansour, 2019a; Cai et al., 2020; Jin et al., 2020a; Shani et al., 2020; Luo et al., 2021; He et al., 2022b) has studied RL with adversarial reward, where the reward is selected by the environment at the beginning of each episode. To cope with adversarial reward, there are generally two iterative schemes. The first scheme is the policy-optimization-based method (Neu et al., 2010; Cai et al., 2020; Luo et al., 2021; He et al., 2022b), where the policy is updated according to the estimated state-value function directly. Following this spirit, under bandit feedback, Neu et al. (2010) achieves a regret upper bound of $\widetilde{O}(T^{2/3})$ with known transition, and Shani et al. (2020) achieves $\widetilde{O}(\sqrt{S^2 A H^4} K^{2/3})$ regret under unknown transition. Under full-information feedback, Cai et al. (2020) establish the first sublinear regret guarantee and POWERS in He et al. (2022b) achieves a near-optimal $\widetilde{O}(dH\sqrt{K})$ regret for adversarial linear mixture MDPs. The second scheme is occupancy-measure-based method (Zimin & Neu, 2013; Rosenberg & Mansour, 2019a;b; Jin et al., 2020a; Luo et al., 2021; Neu & Olkhovskaya,

2021; Dai et al., 2022). The policy is updated under the guidance of optimization of occupancy measure. Guided by this scheme, under known transition, Zimin & Neu (2013) achieves $\widetilde{O}(\sqrt{HSAK})$ regret for bandit feedback and near-optimal $\widetilde{O}(H\sqrt{K})$ regret for full-information feedback. For unknown transition and bandit feedback, an $\widetilde{O}(HS\sqrt{AK})$ regret is achieved by Jin et al. (2020a). Under linear function approximation with bandit feedback, $\widetilde{O}(H\sqrt{dK})$ regret is achieved achieved for linear MDPs (Neu & Olkhovskaya, 2021) and $\widetilde{O}(dS^2\sqrt{K} + \sqrt{HSAK})$ regret is achieved for linear mixture MDPs (Zhao et al., 2023). In this work, we use occupancy-measure-based method to deal with adversarial reward and focus on the setting of linear mixture MDPs[1] with full-information feedback.

## 3 PRELIMINARIES

We study RL for episodic linear mixture MDPs with adversarial reward. We introduce the definitions and necessary assumptions as follows.

### 3.1 MDPs WITH ADVERSARIAL REWARD

We denote a homogeneous, episodic MDP by a tuple $M = M(\mathcal{S}, \mathcal{A}, H, \{r^k\}_{k\in[K]}, \mathbb{P})$, where $\mathcal{S}$ is the state space and $\mathcal{A}$ is the action space, $H$ is the length of the episode, $r^k : \mathcal{S} \times \mathcal{A} \to [0, 1/H]$ is the reward function at the $k$-th episode, and $\mathbb{P}(\cdot|\cdot,\cdot)$ is the transition kernel from a state-action pair to the next state. $r^k$ is adversarially chosen by the environment at the beginning of the $k$-th episode and revealed to the agent at the end of that episode. A policy $\pi = \{\pi_h\}_{h=1}^{H}$ is a collection of functions $\pi_h : \mathcal{S} \to \Delta(\mathcal{A})$.

### 3.2 VALUE FUNCTION AND REGRET

For $(s, a) \in \mathcal{S} \times \mathcal{A}$, we define the action-value function $Q_{k,h}^{\pi}$ and (state) value function $V_{k,h}^{\pi}$ as follows:

$$Q_{k,h}^{\pi}(s,a) = r^k(s,a) + \mathbb{E}\left[\sum\nolimits_{h'=h+1}^{H} r^k(s_{h'}, a_{h'})\middle| s_h = s, a_h = a\right],$$

$$V_{k,h}^{\pi}(s) = \mathbb{E}_{a\sim\pi_h(\cdot|s)}\left[Q_{k,h}^{\pi}(s,a)\right], V_{k,H+1}^{\pi}(s) = 0.$$

Here in the definition of $Q_{k,h}^{\pi}$, we use $\mathbb{E}[\cdot]$ to denote the expectation over the state-action sequences $(s_h, a_h, s_{h+1}, a_{h+1}, .., s_H, a_H)$, where $s_h = s, a_h = a$ and $s_{h'+1} \sim \mathbb{P}(\cdot|s_{h'}, a_{h'})$, $a_{h'+1} \sim \pi_{h'+1}(\cdot|s_{h'+1})$ for all $h' = h, ...H - 1$. For simplicity, for any function $V : \mathcal{S} \to \mathbb{R}$, we denote

$$[\mathbb{P}V](s,a) = \mathbb{E}_{s'\sim\mathbb{P}(\cdot|s,a)}V(s'), \ [\mathbb{V}V](s,a) = [\mathbb{P}V^2](s,a) - \left([\mathbb{P}V](s,a)\right)^2,$$

where $V^2$ is a shorthand for the function whose value at state $s$ is $\left(V(s)\right)^2$. Using this notation, for policy $\pi$, we have the following Bellman equality $Q_{k,h}^{\pi}(s,a) = r^k(s,a) + [\mathbb{P}V_{k,h+1}^{\pi}](s,a)$.

In the online setting, the agent determines a policy $\pi^k$ and start from a fixed state $s_1$ at the beginning of episode $k$. Then at each stage $h \in [H]$, the agent takes an action $a_h \sim \pi_h^k(\cdot|s_h^k)$ and observes the next state $s_{h+1}^k \sim \mathbb{P}(\cdot|s_h^k, a_h^k)$. For the adversarial reward, the goal of RL is to minimize the expected regret, which is the expected loss of the algorithm relative to the best-fixed policy in hindsight (Cesa-Bianchi & Lugosi, 2006). We denote the optimal policy as $\pi^* = \sup_\pi \sum_{k=1}^{K} V_{k,1}^\pi(s_1)$. Thus the expected regret can be written as:

$$\text{Regret}(K) = \sum\nolimits_{k=1}^{K}\left(V_{k,1}^{*}(s_1) - V_{k,1}^{\pi^k}(s_1)\right),$$

where we use $V_{k,1}^{*}(\cdot)$ to denote $V_{k,1}^{\pi^*}(\cdot)$, which is the expected total reward in episode $k$ under the optimal policy.

In this paper, we focus on achieving a horizon-free bound on $\text{Regret}(K)$. Two assumptions are crucial to this end. The first assumption gives a reward shaping reciprocal of the planning horizon length $H$ so as to offset the dependence on $H$ contributed by reward scaling (Jiang & Agarwal, 2018).

**Assumption 3.1** (Dann & Brunskill (2015)). $r^k(s_h, a_h) \leq 1/H, \forall h \in [H]$ for any trajectory $\{s_h, a_h\}_{h=1}^{H}$ induced by $a_h \sim \pi_h(\cdot|s_h)$ and $s_{h+1} \sim \mathbb{P}(\cdot|s_h, a_h)$ for any policy $\pi$ in every episode.

---

[1]Due to limited space, we defer previous works on RL with function approximation to Appendix A.

The intuition behind Assumption 3.1 is the absence of spiky reward in all episodes. The next assumption assumes the transition kernel $\mathbb{P}$ enjoys a linear representation w.r.t. a triplet feature mapping. We define the linear mixture MDPs (Jia et al., 2020; Ayoub et al., 2020; Zhou et al., 2021; Zhou & Gu, 2022) as follows.[2]

**Assumption 3.2** (Linear mixture MDP). A MDP $M = (\mathcal{S}, \mathcal{A}, H, \{r^k\}_{k \in [K]}, \mathbb{P})$ is called an episode $B$-bounded linear mixture MDP, if there exists a *known* feature mapping $\phi(s'|s, a) : \mathcal{S} \times \mathcal{A} \times \mathcal{S} \to \mathbb{R}^d$ and an *unknown* vector $\boldsymbol{\theta}^* \in \mathbb{R}^d$ such that $\mathbb{P}(s'|s, a) = \langle \phi(s'|s, a), \boldsymbol{\theta}^* \rangle$ for any state-action-next-state triplet $(s, a, s')$. We assume $\|\boldsymbol{\theta}^*\|_2 \leq B$ and for any bounded function $V : \mathcal{S} \to [0, 1]$ and any $(s, a) \in \mathcal{S} \times \mathcal{A}$, we have $\|\phi_V(s, a)\|_2 \leq 1$, where $\phi_V(s, a) = \sum_{s' \in \mathcal{S}} \phi(s'|s, a) V(s')$.

Linear mixture MDPs have the following key properties. For any function $V : \mathcal{S} \to \mathbb{R}$ and any state-action pair $(s, a) \in \mathcal{S} \times \mathcal{A}$, the conditional expectation of $V$ over $\mathbb{P}(\cdot|s, a)$ is a linear function of $\boldsymbol{\theta}^*$, i.e., $[\mathbb{P}V](s, a) = \langle \phi_V(s, a), \boldsymbol{\theta}^* \rangle$. Meanwhile, the conditional variance of $V$ over $\mathbb{P}(s, a)$ is quadratic in $\boldsymbol{\theta}^*$, i.e., $[\mathbb{V}V](s, a) = \langle \phi_{V^2}(s, a), \boldsymbol{\theta}^* \rangle - [\langle \phi_V(s, a), \boldsymbol{\theta}^* \rangle]^2$.

### 3.3 Occupancy Measure

We introduce occupancy measure (Altman, 1999; Jin et al., 2020a) as a proxy of the stochastic policy, which will be used in our algorithm design. The occupancy measure $z^\pi = \{z_h^\pi : \mathcal{S} \times \mathcal{A} \times \mathcal{S} \to [0, 1]\}_{h=1}^H$ associated with a stochastic policy $\pi$ and a transition function $\mathbb{P}$ is defined as:

$$z_h^\pi(s, a, s'; \mathbb{P}) = \mathbb{E}[\mathbb{1}\{s_h = s, a_h = a, s_{h+1} = s'\}|\pi, \mathbb{P}].$$

A reasonable occupancy measure $z^\pi$ must satisfy the following constraints:

$$\sum_{s \in \mathcal{S}, a \in \mathcal{A}, s' \in \mathcal{S}} z_h^\pi(s, a, s') = 1. \qquad \text{// Normalization} \qquad (3.1)$$

$$\sum_{a \in \mathcal{A}, y \in \mathcal{S}} z_h^\pi(s, a, y) = \sum_{x \in \mathcal{S}, b \in \mathcal{A}} z_{h-1}^\pi(x, b, s), \forall (s, h) \in \mathcal{S} \times [2 : H]. \quad \text{// Same marginal occupancy}$$

$$(3.2)$$

$$z_1^\pi(s, a, s') = \pi_1(a|s) \, \mathbb{1}\{s = s_1\} \, \mathbb{P}(s'|s, a), \forall (s, a) \in \mathcal{S} \times \mathcal{A}. \qquad \text{// Initial distribution} \quad (3.3)$$

**Lemma 3.3** (Rosenberg & Mansour (2019a)). If a set of functions $z^\pi = \{z_h^\pi : \mathcal{S} \times \mathcal{A} \times \mathcal{S} \to [0, 1]\}_{h=1}^H$ satisfies (3.1) and (3.2), then it is a valid occupancy measure. This occupancy measure is associated with the following induced transition function $\mathbb{P}$, and induced policy $\pi$:

$$\mathbb{P}_h(s'|s, a) = \frac{z_h^\pi(s, a, s')}{\sum_{s'' \in \mathcal{S}} z_h^\pi(s, a, s'')}, \quad \pi_h(a|s) = \frac{\sum_{s' \in \mathcal{S}} z_h^\pi(s, a, s')}{\sum_{a' \in \mathcal{A}, x \in \mathcal{S}} z_h^\pi(s, a', x)}, \qquad (3.4)$$

for all $(s, a, s', h) \in \mathcal{S} \times \mathcal{A} \times \mathcal{S} \times [H]$.

We use $z^*$ to denote the occupancy measure induced by the optimal fixed-policy in hindsight, $\pi^*$ and the true transition function, $\mathbb{P}$.

## 4 The Proposed Algorithm

In this section, we demonstrate HF-O$^2$PS for learning episodic linear mixture MDPs with adversarial reward. At a high level, in each episode, HF-O$^2$PS can be divided into two steps. HF-O$^2$PS firstly updates the policy based on observed data, then uses VTR (Jia et al., 2020; Ayoub et al., 2020) to learn the transition model. To achieve a horizon-free regret, we use occupancy-measure-guided mirror descent rather than proximal policy optimization to update the policy, and adopt variance-uncertainty-aware linear regression and high-order moment estimator . See Section 6 for details.

### 4.1 OMD on occupancy measure

At the beginning of each episode, following Jin et al. (2020a) and Kalagarla et al. (2020), HF-O$^2$PS uses occupancy measures to update the policy based on the observed data. First we calculate the occupancy measure of this episode $\{z_h^k\}_{h=1}^H$ based on the occupancy measure $\{z_h^{k-1}\}_{h=1}^H$ and the reward $\{r_h^{k-1}\}_{h=1}^H$ of the last episode. To utilize learned information, we hope that the transition induced by the new occupancy measure is close to our estimation. Given the confidence set of $\boldsymbol{\theta}^*$

---

[2]We inevitably only consider finite $\mathcal{S}$ and $\mathcal{A}$ due to technical reasons (see Section 5 for details).

---

**Algorithm 1** HF-O$^2$PS

---

**Require:** Regularization parameter $\lambda$, an upper bound $B$ of the $\ell_2$-norm of $\boldsymbol{\theta}^*$, confidence radius $\{\widehat{\beta}_k\}_{k\geq 1}$, level $M$, variance parameters $\xi, \gamma$, $\overline{[M]} = \{0, \ldots, M-1\}$, learning rate $\alpha$

1: Set initial occupancy measure $\{z_h^0(\cdot, \cdot, \cdot)\}_{h=1}^H$ as uniform distribution and assume $r^0(\cdot, \cdot) = 0$.
2: For $m \in \overline{[M]}$, set $\widehat{\boldsymbol{\theta}}_{1,m} \leftarrow \mathbf{0}$, $\widetilde{\boldsymbol{\Sigma}}_{0,H+1,m} \leftarrow \lambda \mathbf{I}$, $\widetilde{\mathbf{b}}_{0,H+1,m} \leftarrow \mathbf{0}$. Set $V_{1,H+1}(\cdot) \leftarrow 0, \mathcal{C}_1 \leftarrow \{\boldsymbol{\theta} : \|\boldsymbol{\theta}\| \leq \beta_1\}$
3: **for** $k = 1, \ldots, K$ **do**
4:   Set $\mathcal{C}_k \leftarrow \{\boldsymbol{\theta} : \|\widehat{\boldsymbol{\Sigma}}_{k,0}^{1/2}(\boldsymbol{\theta} - \widehat{\boldsymbol{\theta}}_{k,0})\|_2 \leq \widehat{\beta}_k\}, \mathcal{D}_k$ as in (4.1)
5:   $\pi^k \leftarrow$ Algorithm $2(z^{k-1}, \mathcal{D}_k, \alpha)$
6:   **for** $h = 1, \ldots, H$ **do**
7:     Take action $a_h^k \sim \pi_h^k(\cdot|s_h^k)$ and receive next state $s_{h+1}^k \sim \mathbb{P}_h(\cdot|s_h^k, a_h^k)$
8:     Observe the adversarial reward function $r^k(\cdot, \cdot)$
9:   **end for**
10:   **for** $h = H, \ldots, 1$ **do**
11:     Set $Q_{k,h}(\cdot, \cdot), V_{k,h}(\cdot)$ as in (4.4) and (4.5)
12:   **end for**
13:   For $m \in \overline{[M]}$, set $\widetilde{\boldsymbol{\Sigma}}_{k,1,m} \leftarrow \widetilde{\boldsymbol{\Sigma}}_{k-1,H+1,m}$
14:   **for** $h = 1, \ldots, H$ **do**
15:     For $m \in \overline{[M]}$, denote $\phi_{k,h,m} = \phi_{V_{k,h+1}^{2^m}}(s_h^k, a_h^k)$.
16:     Set $\{\bar{\sigma}_{k,h,m}\}_{m\in\overline{[M]}} \leftarrow$ Algorithm $3(\{\phi_{k,h,m}, \widehat{\boldsymbol{\theta}}_{k,m}, \widetilde{\boldsymbol{\Sigma}}_{k,h,m}, \widehat{\boldsymbol{\Sigma}}_{k,m}\}_{m\in\overline{[M]}}, \widehat{\beta}_k, \xi, \gamma)$
17:     For $m \in \overline{[M]}$, set $\widetilde{\boldsymbol{\Sigma}}_{k,h+1,m} \leftarrow \widetilde{\boldsymbol{\Sigma}}_{k,h,m} + \phi_{k,h,m}\phi_{k,h,m}^\top/\bar{\sigma}_{k,h,m}^2$
18:     For $m \in \overline{[M]}$, set $\widetilde{\mathbf{b}}_{k,h+1,m} \leftarrow \widetilde{\mathbf{b}}_{k,h,m} + \phi_{k,h,m}V_{k,h+1}^{2^m}(s_{h+1}^k)/\bar{\sigma}_{k,h,m}^2$
19:   **end for**
20:   For $m \in \overline{[M]}$, set $\widehat{\boldsymbol{\Sigma}}_{k+1,m} \leftarrow \widetilde{\boldsymbol{\Sigma}}_{k,H+1,m}, \widehat{\mathbf{b}}_{k+1,m} \leftarrow \widetilde{\mathbf{b}}_{k,H+1,m}, \widehat{\boldsymbol{\theta}}_{k+1,m} \leftarrow \widehat{\boldsymbol{\Sigma}}_{k+1,m}^{-1}\widehat{\mathbf{b}}_{k+1,m}$
21: **end for**

---

**Algorithm 2** Mirror Descent on Occupancy Measure

---

**Require:** the occupancy measure of last iteration $z^{k-1}$, constraint set $\mathcal{D}_k$, learning rate $\alpha$

1: **for** $(h, s, a, s') \in [H] \times \mathcal{S} \times \mathcal{A} \times \mathcal{S}$ **do**
2:   Set $w_h^k(s, a, s') \leftarrow z_h^{k-1}(s, a, s') \exp\{\alpha r_h^{k-1}(s, a)\}$
3:   Set $z^k \leftarrow \arg\min_{z \in \mathcal{D}_k} D_\Phi(z, w^k)$
4:   Set $\pi_h^k(a|s) \leftarrow \frac{\sum_x z_h^k(s,a,x)}{\sum_{a,y} z_h^k(s,a,y)}$
5: **end for**

---

at the beginning of $k$-th episode, $\mathcal{C}_k$ (Line 4, Algorithm 1), we construct the feasible domain of occupancy measure $\mathcal{D}_k$ such that for all occupancy lies in $\mathcal{D}_k$, the transition it induced lies in the confidence set $\mathcal{C}_k$.[3]

**Definition 4.1.** Given the confidence set $\mathcal{C}_k$ of parameter $\boldsymbol{\theta}^*$, we define the feasible occupancy measure set $\mathcal{D}_k \subseteq \mathbb{R}^{|\mathcal{S}|^2|\mathcal{A}|}$ as follows:

$$
\begin{aligned}
\mathcal{D}_k = \Big\{ & z_h(\cdot, \cdot, \cdot) \in \mathbb{R}^{|\mathcal{S}|^2|\mathcal{A}|}, h \in [H] \mid z_h(\cdot, \cdot, \cdot) \geq 0; \\
& \sum_{a,y} z_h(s, a, y) = \sum_{a,y} z_{h-1}(y, a, s), \forall (s, h) \in \mathcal{S} \times [2 : H]; \\
& \sum_{a,s'} z_1(s, a, s') = \mathbf{1}\{s = s_1\}, \forall s \in \mathcal{S}; \forall (s, a, h) \in \mathcal{S} \times \mathcal{A} \times [H], \text{s.t.} \sum_{y\in\mathcal{S}} z_h(s, a, y) > 0, \\
& \exists \bar{\boldsymbol{\theta}}_{s,a,h,k} \in \mathcal{C}_k, \text{s.t.} \frac{z_h(s, a, \cdot)}{\sum_{y\in S} z_h(s, a, y)} = \langle \bar{\boldsymbol{\theta}}_{s,a,h,k}, \phi(\cdot|s, a) \rangle \Big\}.
\end{aligned}
\tag{4.1}
$$

---

[3]We detail the intuition behind Definition 4.1 and computational issues of Line 3 in Appendix E.

---

**Algorithm 3** High-order moment estimator (HOME) (Zhou & Gu, 2022)

---

**Require:** Features $\{\phi_{k,h,m}\}_{m\in[M]}$, vector estimators $\{\widehat{\theta}_{k,m}\}_{m\in[M]}$, covariance matrix $\{\widehat{\Sigma}_{k,m}\}_{m\in[M]}$ and $\{\widetilde{\Sigma}_{k,h,m}\}_{m\in[M]}$, confidence radius $\widehat{\beta}_k, \xi, \gamma$

1: **for** $m = 0, \dots, M-2$ **do**
2:     Set $[\bar{\mathbb{V}}_{k,m}V_{k,h+1}^{2^m}](s_h^k, a_h^k) \leftarrow [\langle\phi_{k,h,m+1}, \widehat{\theta}_{k,m+1}\rangle]_{[0,1]} - [\langle\phi_{k,h,m}, \widehat{\theta}_{k,m}\rangle]_{[0,1]}^2$
3:     Set $E_{k,h,m} \leftarrow \min\{1, 2\widehat{\beta}_k\|\phi_{k,h,m}\|_{\widehat{\Sigma}_{k,m}^{-1}}\} + \min\{1, \widehat{\beta}_k\|\phi_{k,h,m+1}\|_{\widehat{\Sigma}_{k,m+1}^{-1}}\}$
4:     Set $\bar{\sigma}_{k,h,m}^2 \leftarrow \max\{[\bar{\mathbb{V}}_{k,m}V_{k,h+1}^{2^m}](s_h^k, a_h^k) + E_{k,h,m}, \xi^2, \gamma^2\|\phi_{k,h,m}\|_{\widetilde{\Sigma}_{k,h,m}^{-1}}\}$
5: **end for**
6: Set $\bar{\sigma}_{k,h,M-1}^2 \leftarrow \max\{1, \xi^2, \gamma^2\|\phi_{k,h,M-1}\|_{\widetilde{\Sigma}_{k,h,M-1}^{-1}}\}$

**Ensure:** $\{\bar{\sigma}_{k,h,m}\}_{m\in[M]}$

---

We define the standard mirror map $\Phi$ for the probability simplex $z$ and the corresponding Bregman divergence $D_\Phi$ as follows:

$$\Phi(z) = \sum_{h=1}^{H}\sum_{s,a,s'} z_h(s,a,s')(\log z_h(s,a,s')-1), \quad D_\Phi(x,y) = \Phi(x)-\Phi(y)-\langle x-y, \nabla\Phi(y)\rangle. \quad (4.2)$$

And the following lemma shows that our mirror map is $1/H$-strongly convex.

**Lemma 4.2.** $\Phi$ is $1/H$-strongly convex on the space of occupancy measure with respect to $\|\cdot\|_1$, thus strongly convex on $\mathcal{D}_k$.

The basic idea of updating $z^k$ is to minimize the trade-off between the value-loss and the distance from the occupancy measure of last episode. Formally we have:

$$z^k = \arg\min_{z\in\mathcal{D}_k} \alpha\langle z^{k-1}, r^{k-1}\rangle + D_\Phi(z, z^{k-1}), \quad (4.3)$$

where $\alpha$ is the learning rate and the inner product is defined as follows:

$$\langle z, r\rangle = \sum_{s,a,s',h\in\mathcal{S}\times\mathcal{A}\times\mathcal{S}\times[H]} z_h(s,a,s')r(s,a).$$

Following Rosenberg & Mansour (2019a), we split (4.3) to the two-step optimization at Line 2 and 3 of Algorithm 2. By Lemma 3.3, we update the policy as follows:

$$\pi_h^k = \frac{\sum_{s'} z_h^k(s,a,s')}{\sum_{a,s'} z_h^k(s,a,s')}.$$

For sake of simplicity, we also define the optimistic expected total reward given by the occupancy measure $\bar{V}_{k,1}(s_1)$ as follows:

$$\bar{V}_{k,1}(s_1) = \sum_{h,s,a,a'} z_h^k(s,a,s')r(s,a)$$

After obtaining $\pi^k$, HF-O$^2$PS chooses actions $a_h^k$ based on our new policy $\pi_h^k$ and observe the whole reward function $r^k$ at the end of the episode.

### 4.2 VTR WITH HIGH-ORDER MOMENT ESTIMATION

The second phase of HF-O$^2$PS is to estimate the transition model $\langle\theta^*, \phi\rangle$ and evaluate the policy $\pi^k$. In this step, we construct a variance-uncertainty-aware weighted least square estimator (Zhou & Gu, 2022) and explicitly estimate higher moments of $\mathbb{P}$ (Zhang et al., 2021b; Zhou & Gu, 2022), which are poly($\theta^*$) under Assumption 3.2.

Concretely, we first compute the optimistic estimation of $Q_h^{\pi^k}$ (resp. $V_h^{\pi^k}$), $Q_{k,h}$ (resp. $V_{k,h}$), in a backward manner. Specifically, HF-O$^2$PS computes the optimistic $Q_{k,h}$ and $V_{k,h}$ as:

$$Q_{k,h}(\cdot,\cdot) = \left[r^k(\cdot,\cdot) + \langle\widehat{\theta}_{k,0}, \phi_{V_{k,h+1}}(\cdot,\cdot)\rangle + \widehat{\beta}_k\|\phi_{V_{k,h+1}}(\cdot,\cdot)\|_{\widehat{\Sigma}_{k,0}^{-1}}\right]_{[0,1]}, \quad (4.4)$$

$$V_{k,h}(\cdot) = \mathbb{E}_{a\sim\pi_h^k(\cdot|\cdot)}[Q_{k,h}(\cdot,a)], \quad (4.5)$$

where $\widehat{\boldsymbol{\theta}}_{k,0}$ is the 0-th estimator of $\boldsymbol{\theta}^*$, $\widehat{\boldsymbol{\Sigma}}_{k,0}$ is the covariance matrix and $\widehat{\beta}_k$ is the radius of the confidence set defined as:

$$\widehat{\beta}_k = 12\sqrt{d\log(1 + kH/(\xi^2 d\lambda))\log(32(1+\log(\gamma^2/\xi)))k^2H^2/\delta)}$$
$$+ 30\log(32(1+\log(\gamma^2/\xi))k^2H^2/\delta)/\gamma^2 + \sqrt{\lambda}B, \tag{4.6}$$

Then we estimate $\boldsymbol{\theta}^*$ by a weighted regression problem with predictor $\boldsymbol{\phi}_{k,h,0} = \boldsymbol{\phi}_{V_{k,h+1}}(s_h^k, a_h^k)$ against response $V_{k,h+1}(s_{h+1}^k)$. Specifically, $\widehat{\boldsymbol{\theta}}_{k,0}$ is the solution to the VTR problem:

$$\operatorname*{argmin}_{\boldsymbol{\theta}} \lambda\|\boldsymbol{\theta}\|_2^2 + \sum_{j=1}^{k-1}\sum_{h=1}^{H}[\langle\boldsymbol{\phi}_{j,h,0},\boldsymbol{\theta}\rangle - V_{j,h+1}(s_{h+1}^j)]^2/\bar{\sigma}_{j,h,0}^2,$$

where the weight $\bar{\sigma}_{j,h,0}^2$ is a high-probability upper bound of the conditional variance $[\mathbb{V}V_{j,h+1}](s_h^j, a_h^j)$. In detail, for each $k \in [K]$ and $a \in \mathcal{A}$, if $[\mathbb{V}V_{k,h+1}](s_h^k, a_h^k)$ can be computed for a function $V$ efficiently, we define

$$\bar{\sigma}_{k,h,0}^2 = \max\{[\mathbb{V}V_{k,h+1}](s_h^k, a_h^k), \xi^2, \gamma^2\|\boldsymbol{\phi}_{k,h,0}\|_{\widetilde{\boldsymbol{\Sigma}}_{k,h,0}^{-1}}\}, \tag{4.7}$$

where $[\mathbb{V}V_{k,h+1}](s_h^k, a_h^k)$ is the *variance-aware* term and $\gamma^2\|\boldsymbol{\phi}_{k,h,0}\|_{\widetilde{\boldsymbol{\Sigma}}_{k,h,0}^{-1}}$ is the *uncertainty-aware* term.

However, we choose to replace $[\mathbb{V}V_{k,h+1}](s_h^k, a_h^k)$ with $[\bar{\mathbb{V}}V_{k,h+1}](s_h^k, a_h^k) + E_{k,h,0}$ in (4.7) since the true transition $\mathbb{P}$ is unknown, and hence the true conditional variance is not exactly available. Here $E_{k,h,0}$ (Line 3 in Algorithm 3) is an error bound such that $[\bar{\mathbb{V}}V_{k,h+1}](s_h^k, a_h^k) + E_{k,h,0} \geq [\mathbb{V}V_{k,h+1}](s_h^k, a_h^k)$ with high probability and $[\bar{\mathbb{V}}V_{k,h+1}](s_h^k, a_h^k)$ (Line 2 in Algorithm 3) is designed as

$$[\langle\boldsymbol{\phi}_{k,h,1}, \widehat{\boldsymbol{\theta}}_{k,1}\rangle]_{[0,1]} - [\langle\boldsymbol{\phi}_{k,h,0}, \widehat{\boldsymbol{\theta}}_{k,0}\rangle]_{[0,1]}^2,$$

where $\boldsymbol{\phi}_{k,h,1} = \boldsymbol{\phi}_{V_{k,h+1}^2}(s_h^k, a_h^k)$ and $\widehat{\boldsymbol{\theta}}_{k,1}$ is the solution to the $\bar{\sigma}_{k,h,1}^2$-weighted regression problem with predictor $\boldsymbol{\phi}_{k,h,1}$ against response $V_{k,h+1}^2(s_{h+1}^k)$. Similar to the estimating procedure of $\widehat{\boldsymbol{\theta}}_{k,0}$, we set $\bar{\sigma}_{k,h,1}^2$ based on $[\bar{\mathbb{V}}V_{k,h+1}^2](s_h^k, a_h^k) + E_{k,h,1}$, which is an upper bound of $[\mathbb{V}V_{k,h+1}^2](s_h^k, a_h^k)$ with high probability. Repeating this process, we recursively estimate the conditional $2^m$-th moment of $V_{k,h+1}$ by its variance in Algorithm 3, which is dubbed as *high-order moment estimator*.

## 5 MAIN RESULTS

### 5.1 REGRET UPPER BOUND FOR HF-O$^2$PS

We first provide the regret bound for HF-O$^2$PS.

**Theorem 5.1.** Set $M = \log_2(4KH)$, $\xi = \sqrt{d/(KH)}$, $\gamma = 1/d^{1/4}$, $\lambda = d/B^2$, and $\alpha = H/\sqrt{K}$. For any $\delta > 0$, with probability at least $1 - (3M + 2)\delta$, Algorithm 1 yields a regret bounded as follows:

$$\mathrm{Regret}(K) = \widetilde{O}\Big(\big(d + \log\big(|\mathcal{S}|^2|\mathcal{A}|\big)\big)\sqrt{K} + d^2\Big). \tag{5.1}$$

**Remark 5.2.** By omitting the logarithmic terms in (5.1), HF-O$^2$PS achieves a horizon free regret upper bound $\widetilde{O}(d\sqrt{K} + d^2)$. Our regret bound is better than $\widetilde{O}((H + d)\sqrt{K} + d^2 H)$ obtained by He et al. (2022b) when $H = \Omega(\log|\mathcal{S}|)$. Additionally, compared with HF-UCRL-VTR+ algorithm proposed by Zhou & Gu (2022) for episodic linear mixture MDPs with fixed reward, HF-O$^2$PS provides a robustness against adversarial reward while maintaining its regret upper bounded by $\widetilde{O}(d\sqrt{K} + d^2)$.

### 5.2 HARDNESS RESULTS

We also provide two regret lower bounds. The next theorem gives a regret lower bound of MDPs with known transition and adversarial reward.

**Theorem 5.3.** When $H = 2\widetilde{H}$, where $\widetilde{H}$ is a positive integer, for any algorithm and any given nonempty action space $\mathcal{A}$, there exists an MDP satisfying Assumptions 3.1 and 3.2 with $d = 1$ and $|\mathcal{S}| = \Theta(|\mathcal{A}|^H)$ such that

$$\lim_{\widetilde{H}\to\infty}\lim_{K\to\infty}\frac{\mathbb{E}[\mathrm{Regret}(K)]}{\sqrt{HK\log|\mathcal{A}|}} \geq c_1 = \frac{1}{\sqrt{2}} \quad \text{and} \quad \lim_{\widetilde{H}\to\infty}\lim_{K\to\infty}\frac{\mathbb{E}[\mathrm{Regret}(K)]}{\sqrt{K\log|\mathcal{S}|}} \geq c_2 = \frac{1}{2\sqrt{2}}.$$

**Remark 5.4.** Theorem 5.3 indicates that even the estimation error $I_2$ disappears in (6.1), which means we are in the "learning-free" setting, with infinitely large $\mathcal{S}$, purely the adversarial environment can introduce a $\sqrt{H}$ dependency asymptotically. Therefore, we can only expect a horizon-free algorithm whose regret upper bound at least depends on $\log |\mathcal{S}|$, and $K$.

The following theorem provides another regret lower bound of learning homogeneous linear mixture MDPs with adversarial reward.

**Theorem 5.5.** Let $B > 1$ and $K > \max\{3d^2, (d-1)/(192(b-1))\}$, for any algorithm. there exists a $B$-bounded adversarial MDP satisfying Assumption 3.1and  3.2, such that the expected regret $\mathbb{E}[\text{Regret}(K)]$ has lower bound $d\sqrt{K}/(16\sqrt{3})$.

**Remark 5.6.** Theorem 5.5 shows that when $K$ is large enough, any algorithm for adversarial MDPs satisfying Assumption 3.1 and  3.2 has regret at least $\Omega(d\sqrt{K})$. Moreover, the regret lower bound in Theorem 5.5 matches the regret upper bound in Theorem 5.1, which suggests that HF-O$^2$PS is near-optimal.

## 6 PROOF OVERVIEW

In this section, we provide the proof sketch of Theorem 5.1 and illustrate the key technical issues.

*Proof sketch of Theorem 5.1.* First, we have the regret decomposition:

$$\text{Regret}(K) = \underbrace{\sum_{k=1}^{K} \left(V_{k,1}^*(s_1) - \bar{V}_{k,1}(s_1)\right)}_{I_1} + \underbrace{\sum_{k=1}^{K} \left(V_{k,1}(s_1) - V_1^{\pi_k}(s_1)\right)}_{I_2} + \underbrace{\sum_{k=1}^{K} \left(\bar{V}_{k,1}(s_1) - V_{k,1}(s_1)\right)}_{I_3}.$$

(6.1)

**Bounding $I_1$.** $I_1$ is the regret of policy updating. By the standard regret analysis of OMD, the regret on probability simplex is bounded by $\widetilde{O}(L\sqrt{K})$ where $L$ is the upper bound of the gradients and $K$ is the number of iterations. In MDPs, we have $H$ decisions to make in each episode. Therefore, policy updating can be seen as conducting mirror descent simultaneously on $H$ simplexes, and the total regret is the summation of regret on each simplexes. Consequently, the regret upper bound is roughly $\widetilde{O}(H\bar{L}\sqrt{K})$, where $\bar{L}$ is the average upper bound of the gradients over all the simplexes. In OPPO (Cai et al., 2020) and POWERS (He et al., 2022b), the policy is updated via proximal policy optimization: $\pi_h^k(a|s) \propto \pi_h^{k-1}(a|s) \exp\{\alpha Q_{k-1,h}(s,a)\}$. Hence the gradients is $Q_{k-1,h}(s,a)$, which, after taking average over $h \in [H]$, result in an average $\bar{L} = O(1)$ and consequently a regret bound of $\widetilde{O}(H\sqrt{K})$. To address this issue, we consider using $r^k$ as the gradients, which is enabled by introducing an occupancy measure. By Assumption 3.1, the standard regret analysis of OMD results in $I_1 = \widetilde{O}(\sqrt{K})$.

**Bounding $I_2$.** $I_2$ can be further decomposed into three major terms, the sum of bonus, transition noise and policy noise. Roughly, we have:

$$I_2 = \underbrace{\sum_{k=1}^{K}\sum_{h=2}^{H}[\mathbb{P}V_{k,h}(s_{h-1}^k, a_{h-1}^k) - V_{k,h}(s_h^k)]}_{\text{(ii) transition noise}} + \underbrace{\sum_{k=1}^{K}\sum_{h=2}^{H}[\mathbb{E}_{a\sim\pi_h^k(\cdot|s_h^k)}[Q_{k,h}(s_h^k, a)] - Q_{k,h}(s_h^k, a_h^k)]}_{\text{(iii) policy noise}}$$

$$+ \underbrace{\sum_{k=1}^{K}\sum_{h=1}^{H}[Q_{k,h}(s_h^k, a_h^k) - r(s_h^k, a_h^k) - \mathbb{P}V_{k,h+1}(s_h^k, a_h^k)]}_{\text{bonus terms}} + \Gamma,$$

where $\Gamma$ is defined as follows, which can be bounded by $\widetilde{O}(\sqrt{K})$ using Azuma-Hoeffding's inequality:

$$\Gamma = \sum_{k=1}^{K} \left(\mathbb{E}_{a\sim\pi_1^k(\cdot|s_1^k)}[Q_{k,1}(s_1^k, a)|s_1^k] - Q_{k,1}(s_1^k, a_1^k)\right) + \sum_{k=1}^{K} \left(\sum_{h=1}^{H} r(s_h^k, a_h^k) - V_1^{\pi^k}(s_1^k)\right).$$

The standard way to bound the bonus term is applying the total variance lemma (Jin et al., 2018, Lemma C.5) to the total variance of transition noise (He et al., 2022b; Zhou et al., 2021). However,

in our case, a naive adaptation of He et al. (2022b, Lemma 6.4) and total variance lemma results in an upper bound with $\sqrt{KH}$-dependence. Also, the transition noise and policy noise can only be bounded using standard concentration inequalities, which results in another $\sqrt{KH}$ term.

To shave $\text{poly}(H)$ off, we propose to bound the bonus term and transition noise recursively inspired by Zhou & Gu (2022), equipped with the variance-uncertainty-aware weighting mechanism and Algorithm 3. One notable challenge is to tackle the randomness in $\pi^k(\cdot|\cdot)$, (iii), to which simply applying Azuma-Hoeffding's inequality will yield $\widetilde{O}(\sqrt{KH})$. We follow the procedure of bounding (ii) in Zhou & Gu (2022), where the transition noise of order $m$ is first bounded the sum of conditional variance $\mathbb{V}V_{k,h}^{2^m}(s_{h-1}^k, a_{h-1}^k)$ using martingale concentration inequality. Then, the key step is bounding the conditional variance with higher order transition noise as follows:

$$\mathbb{V}V_{k,h}^{2^m}(s_{h-1}^k, a_{h-1}^k) \leq X(m) + \underbrace{\mathbb{P}V_{k,h}^{2^{m+1}}(s_{h-1}^k, a_{h-1}^k) - V_{k,h}^{2^{m+1}}(s_h^k)}_{\text{transition noise of higher order}} + \underbrace{V_{k,h}^{2^{m+1}}(s_h^k) - Q_{k,h}^{2^{m+1}}(s_h^k, a_h^k)}_{(*)},$$

(6.2)

where $X(m)$ only depends on $m$, the second term of the right hand side is exactly the transition noise of higher-order Value function. For $\text{argmax}$ policy, the martingale difference (*) in (6.2) is 0, which indicates that the total variance can be bounded by the martingale difference of higher order. For policy noise which did not appear in (Zhou & Gu, 2022), we first bound (*) by the sum of conditional variance, each term of which is:

$$\mathbb{E}_{a \sim \pi_h^k(\cdot|s_h^k)}[Q_{k,h}^{2^{m+1}}(s_h^k, a)] - \mathbb{E}_{a \sim \pi_h^k(\cdot|s_h^k)}^2[Q_{k,h}^{2^m}(s_h^k, a)]$$
$$= \mathbb{E}_{a \sim \pi_h^k(\cdot|s_h^k)}[Q_{k,h}^{2^{m+1}}(s_h^k, a)] - Q_{k,h}^{2^{m+1}}(s_h^k, a_h^k) + Q_{k,h}^{2^{m+1}}(s_h^k, a_h^k) - \mathbb{E}_{a \sim \pi_h^k(\cdot|s_h^k)}^2[Q_{k,h}^{2^m}(s_h^k, a)].$$

(6.3)

Because (6.2) always holds whether $\pi$ is stochastic, combining (6.2) with (6.3), we have the follows:

$$\mathbb{E}_{a \sim \pi_h^k(\cdot|s_h^k)}[Q_{k,h}^{2^{m+1}}(s_h^k, a)] - \mathbb{E}_{a \sim \pi_h^k(\cdot|s_h^k)}^2[Q_{k,h}^{2^m}(s_h^k, a)] + \mathbb{V}V_{k,h}^{2^m}(s_{h-1}^k, a_{h-1}^k)$$
$$\leq X(m) + \underbrace{\mathbb{P}V_{k,h}^{2^{m+1}}(s_{h-1}^k, a_{h-1}^k) - V_{k,h}^{2^{m+1}}(s_h^k)}_{(\star)} + \underbrace{\mathbb{E}_{a \sim \pi_h^k(\cdot|s_h^k)}[Q_{k,h}^{2^{m+1}}(s_h^k, a)] - Q_{k,h}^{2^{m+1}}(s_h^k, a_h^k)}_{(*)}$$
$$+ \underbrace{V_{k,h}^{2^{m+1}}(s_h^k) - \mathbb{E}_{a \sim \pi_h^k(\cdot|s_h^k)}^2[Q_{k,h}^{2^m}(s_h^k, a)]}_{:=(**)\leq 0},$$

which is nearly the same as (6.2) except (**). Therefore if we view the transition noise $(\star)$ and policy noise (*) as a single martingale, then it can be bounded by total noise of higher order the same as (6.2). The rest framework of HOME in Zhou & Gu (2022) can be adapted smoothly and yields an upper bound $\widetilde{O}(d\sqrt{K} + d^2)$.

**Bounding $I_3$.** $I_3$ is the gap between the optimistic value function derived from occupancy measure guided policy updating and the other one derived from backward iteration (Line 11 of Algorithm 1). By Lemma 3.3, for each $k \in [K]$, the occupancy measure $\{z_h^k\}_{h=1}^H$ induces a new MDP and policy. Then $z^k \in \mathcal{D}_k$ implies that the transition still lies in the confidence set, thus can also be bounded by $Q_{k,h}(\cdot, \cdot)$ and $V_{k,h}(\cdot)$. Formally, we have the following lemma:

**Lemma 6.1.** For all $k \in [K]$, let $\bar{V}_{k,1}(s_1)$ be the optimistic value function given by occupancy measure and $V_{k,1}(s_1)$ the value function computed by backward iteration (Line 11). We have $\bar{V}_{k,1}(s_1) \leq V_{k,1}(s_1)$, and thus $I_3 \leq 0$.

Finally, combining the upper bounds of all three terms finishes our proof. □

## 7 CONCLUSION

In this work, we considered learning homogeneous linear mixture MDPs with adversarial reward. We proposed a new algorithm based on occupancy measure and high-order moment estimator. We show that HF-O$^2$PS achieves the near-optimal regret upper bounded $\widetilde{O}(d\sqrt{K} + d^2)$. To the best of our knowledge, our algorithm is the first horizon-free algorithm in this setting. Currently, our result requires the uniformly bounded reward assumption, i.e., Assumption 3.1. For horizon-free algorithms require only the total reward in each episode bounded by 1, we leave it as future work.

ACKNOWLEDGEMENTS

We thank the anonymous reviewers and area chair for their helpful comments. JH and QG are supported in part by the National Science Foundation CAREER Award 1906169 and Amazon Research Award. JH is also supported in part by Amazon PhD Fellowship. WZ is supported by UCLA Dissertation Year Fellowship. The views and conclusions contained in this paper are those of the authors and should not be interpreted as representing any funding agencies.

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

## A    ADDITIONAL RELATED WORKS

Table 1: A comparison of related horizon-free algorithms and algorithms for adversarial linear mixture MDPs

| Algorithm | Regret | Assumption | Adversarial Reward | Extra Requirements |
|---|---|---|---|---|
| VARLin (Zhang et al., 2021b) | $\widetilde{O}(d^{4.5}\sqrt{K} + d^9)$ | Homogeneous, $\sum_h r_h \leq 1$ | No | - |
| VARLin2 (Kim et al., 2022) | $\widetilde{O}(d\sqrt{K} + d^2)$ | Homogeneous, $\sum_h r_h \leq 1$ | No | - |
| HF-UCRL-VTR$^+$ (Zhou & Gu, 2022) | $\widetilde{O}(d\sqrt{K} + d^2)$ | Homogeneous, $\sum_h r_h \leq 1$ | No | - |
| Lower bound (Zhou & Gu, 2022) | $\Omega(d\sqrt{K})$ | Homogeneous $r_h \leq 1/H$ | No | $K \gtrsim d^2$ |
| OPPO (Cai et al., 2020) | $\widetilde{O}(dH^2\sqrt{K})$ | Inhomogeneous, $\sum_h r_h \leq H$ | Yes | $d \gtrsim \log|\mathcal{A}|,$ $K \gtrsim \text{poly}(d, H)$ |
| POWERS (He et al., 2022b) | $\widetilde{O}(d\sqrt{H^3 K})$ | Inhomogeneous, $\sum_h r_h \leq H$ | Yes | $d \gtrsim \log|\mathcal{A}|,$ $K \gtrsim \text{poly}(d, H)$ |
| **Ours** | **Theorem 5.1** | Homogeneous, $r_h \leq 1/H$ | Yes | $d \gtrsim \log|\mathcal{A}|$ |
| Lower Bound | **Open** At least $\widetilde{\Omega}(d\sqrt{K})$ | Homogeneous, $r_h \leq 1/H$ | Yes | $K \gtrsim d^2$ |

**RL with linear function approximation**    To make MDPs with large state space amenable for provable RL, there has been an explosion of works relying on MDP classes with various linear structures (Jiang et al., 2017; Sun et al., 2019; Du et al., 2021; Jin et al., 2021). Among different assumptions made in recent work (Yang & Wang, 2019; Wang et al., 2020b; Jin et al., 2020b; Du et al., 2019; Zanette et al., 2020; Ayoub et al., 2020; Jia et al., 2020; Weisz et al., 2021; Zhou et al., 2021; He et al., 2022b; Zhou & Gu, 2022; He et al., 2022a), we consider the linear mixture MDP setting (Jia et al., 2020; Ayoub et al., 2020; Zhou et al., 2021; Zhang et al., 2021a; He et al., 2022b), where the transition kernel is a linear combination of $d$ given models. More specifically, we focus on the adversarial linear mixture MDP of He et al. (2022b), whose approach is nearly minimax optimal but insufficient to obtain horizon-free regret, with a refined reward assumption. There is also a parallel line of work (Jin et al., 2020b; He et al., 2022a) investigating the linear MDP model of Jin et al. (2020b) with much larger degree of freedom, where the transition function and reward function are linear in a known state-action feature mapping respectively.

## B    PROOF OF LEMMAS IN SECTION 4 AN SECTION 6

### B.1    PROOF OF LEMMA 4.2

*Proof.* Say we have two occupancy measures $z, w$, then we have

$$
\begin{aligned}
D_\Phi(z||w) &= \sum_{h=1}^{H} \sum_{s,a,s'} z_h(s,a,s') \log \frac{z_h(s,a,s')}{w_h(s,a,s')} \\
&\geq \frac{1}{2} \sum_{h=1}^{H} \left( \sum_{s,a,s'} \left| z_h(s,a,s') - w_h(s,a,s') \right| \right)^2 \\
&\geq \frac{1}{2H} \left( \sum_{h=1}^{H} \sum_{s,a,s'} \left| z_h(s,a,s') - w_h(s,a,s') \right| \right)^2 \\
&= \frac{1}{2H} \|z - w\|_1^2,
\end{aligned}
$$

where the first inequality holds due to Pinsker's inequality and the second inequality holds due to Cauchy-Schwartz inequality.    □

## B.2 Proof of Lemma 6.1

*Proof of Lemma 6.1.* Given a set of occupancy measure, we define the respective transition as the follows:

$$\bar{p}_h^k(s'|s,a) = \langle \bar{\boldsymbol{\theta}}_{s,a,h,k}, \boldsymbol{\phi}(s'|s,a) \rangle = \frac{z_h^k(s,a,s')}{\sum_{s'} z_h^k(s,a,s')},$$

$$\forall (s,a,h) \in \mathcal{S} \times \mathcal{A} \times [H], \text{s.t.} \sum_{s'} z_h^k(s,a,s') > 0.$$

Now let's consider another MDP $M_k' = (\mathcal{S}, \mathcal{A}, H, \{r_h\}, \{\mathbb{P}_{k,h,s,a}\})$, where the state space, action space, length of horizon, reward functions are the same as the true MDP $M$, and $\mathbb{P}_{k,h,s,a}(\cdot|\cdot,\cdot) = \bar{p}_h^k(\cdot|\cdot,\cdot)$. However, our new MDP is a tabular one and its transition kernel is different from $M$. Consider running first inner loop in our algorithm (line 10 - line 12), since $M$ and $M_k'$ share the same reward function, and the other terms also do not depend on true transition, the results running on the two MDPs should be the same.

For the sake of simplicity, we (recursively) define the value functions on the imaginary MDP $M_k'$:

$$\bar{V}_{k,H+1}(s) = 0,$$
$$\bar{Q}_{k,h}(s,a) = r_h(s,a) + \langle \bar{\boldsymbol{\theta}}_{s,a,h,k}, \boldsymbol{\phi}_{\bar{V}_{k,h+1}}(s,a) \rangle,$$
$$\bar{V}_{k,h}(s) = \mathbb{E}_{a \sim \pi_h^k(a|s)}[Q_{k,h}(s,a)].$$

Then it is easy to verify that $\bar{V}_{k,1}(s_1)$ computed by occupancy measure is the same as the one computed by the above way. Then, we can prove our theorem by induction. The conclusion trivially holds for $n = H + 1$. Suppose the statement holds for $n = h + 1$, then for $n = h$, for each $(s,a)$, since $\bar{Q}_{k,h}(s,a) \leq 1$, so if $Q_{k,h}(s,a) = 1$ then the proof is finished. Otherwise we have:

$$Q_{k,h}(s,a) - \bar{Q}_{k,h}(s,a) \geq \langle \widehat{\boldsymbol{\theta}}_{k,0}, \boldsymbol{\phi}_{V_{k,h+1}}(\cdot,\cdot) \rangle + \widehat{\beta}_k \|\boldsymbol{\phi}_{V_{k,h+1}}(\cdot,\cdot)\|_{\widehat{\boldsymbol{\Sigma}}_{k,0}^{-1}} - \langle \bar{\boldsymbol{\theta}}_{s,a,h,k}, \boldsymbol{\phi}_{V_{k,h+1}}(s,a) \rangle$$

$$= \langle \widehat{\boldsymbol{\theta}}_{k,0} - \bar{\boldsymbol{\theta}}_{s,a,h,k}, \boldsymbol{\phi}_{V_{k,h+1}}(\cdot,\cdot) \rangle + \widehat{\beta}_k \|\boldsymbol{\phi}_{V_{k,h+1}}(\cdot,\cdot)\|_{\widehat{\boldsymbol{\Sigma}}_{k,0}^{-1}}$$

$$\geq \widehat{\beta}_k \|\boldsymbol{\phi}_{V_{k,h+1}}(\cdot,\cdot)\|_{\widehat{\boldsymbol{\Sigma}}_{k,0}^{-1}} - \|(\widehat{\boldsymbol{\theta}}_{k,0} - \bar{\boldsymbol{\theta}}_{s,a,h,k})\|_{\widehat{\boldsymbol{\Sigma}}_{k,0}} \|\boldsymbol{\phi}_{V_{k,h+1}}(\cdot,\cdot)\|_{\widehat{\boldsymbol{\Sigma}}_{k,0}^{-1}}$$

$$\geq 0,$$

where the first inequality holds by the inductive hypothesis, the second inequality holds due to Cauchy-Schwartz inequality and the third inequality holds due to $\bar{\boldsymbol{\theta}}_{s,a,h,k} \in \mathcal{C}_k$. By induction, we finish the proof. □

## C Proof of the Main Result

In this section, we are going to provide the proof of Theorem 5.1. First, we define the $\sigma$-algebra generated by the random variables representing the transition noise and the stochastic policy noise. For $k \in [K], h \in [H]$, we define $\mathcal{F}_{k,h}$ the $\sigma$-algebra of state and actions till stage $k$ and step $h$, and $\mathcal{G}_{k,h}$ the state till stage $k$ and step $h$. That is,

$$s_1^1, a_1^1, ..., s_h^1, a_h^1, ..., s_H^1, a_H^1,$$
$$s_1^2, a_1^2, ..., s_h^2, a_h^2, ..., s_H^2, a_H^2,$$
$$...$$
$$s_1^k, a_1^k, ..., s_h^k, a_h^k,$$

generates $\mathcal{F}_{k,h}$, and

$$s_1^1, a_1^1, ..., s_h^1, a_h^1, ..., s_H^1, a_H^1,$$
$$s_1^2, a_1^2, ..., s_h^2, a_h^2, ..., s_H^2, a_H^2,$$
$$...$$
$$s_1^k, a_1^k, ..., s_h^k,$$

generates $\mathcal{G}_{k,h}$. Second, we define $\mathbb{J}_h^k$ as

$$\mathbb{J}_h^k f(s) = \mathbb{E}_{a \sim \pi_h^k(\cdot|s)}[f(s,a)|s], \tag{C.1}$$

for any $(k,h) \in [K] \times [H]$ and function $f : \mathcal{S} \times \mathcal{A} \to \mathbb{R}$ for simplicity.

## C.1 LEMMAS FOR SELF-CONCENTRATION MARTINGALES

In this section, we provide two results of self-concentration martingales, which are key to our proof.

**Lemma C.1** (Lemma B.1, Zhou & Gu 2022). Let $\{\sigma_k, \beta_k\}_{k \geq 1}$ be a sequence of non-negative numbers, $\xi, \gamma > 0$, $\{\mathbf{x}_k\}_{k \geq 1} \subset \mathbb{R}^d$ and $\|\mathbf{x}_k\|_2 \leq L$. Let $\{\mathbf{Z}_k\}_{k \geq 1}$ and $\{\bar{\sigma}_k\}_{k \geq 1}$ be inductively defined in the following way: $Z_1 = \lambda \mathbf{I}$,

$$\forall k \geq 1, \bar{\sigma}_k = \max\{\sigma_k, \xi, \gamma \|\mathbf{x}_k\|_{\mathbf{Z}_k^{-1}}^{1/2}\}, \mathbf{Z}_{k+1} = \mathbf{Z}_k + \mathbf{x}_k \mathbf{x}_k^\top / \bar{\sigma}_k^2.$$

Let $\iota = \log(1 + KL^2/(d\lambda\xi^2))$. Then we have

$$\sum_{k=1}^{K} \min\left\{1, \beta_k \|\mathbf{x}_k\|_{\mathbf{Z}_k^{-1}}\right\} \leq 2d\iota + 2 \max_{k \in [K]} \beta_k \gamma^2 d\iota + 2\sqrt{d\iota}\sqrt{\sum_{k=1}^{K} \beta^2 (\sigma^2 + \xi^2)}.$$

Same as in Zhou & Gu (2022), first we need to prove that the vector $\boldsymbol{\theta}^*$ lies in the series of confidence sets, which implies the estimation we get via occupancy measure is optimistic and the high-order moments are close to their true values.

**Lemma C.2** (Lemma C.1, Zhou & Gu 2022). Set $\{\widehat{\beta}_k\}_{k \geq 1}$ as (4.6), then, with probability at least $1 - M\delta$, we have for any $k \in [K], h \in [H], m \in \overline{[M]}$,

$$\left\|\left(\widehat{\boldsymbol{\theta}}_{k,m} - \boldsymbol{\theta}^*\right)\right\|_{\widehat{\boldsymbol{\Sigma}}_{k,m}} \leq \widehat{\beta}_k, \left|[\bar{\mathbb{V}}_{k,m} V_{k,h+1}^{2^m}](s_h^k, a_h^k) - [\mathbb{V} V_{k,h+1}^{2^m}](s_h^k, a_h^k)\right| \leq E_{k,h,m}.$$

Let $\mathcal{E}_{C.2}$ denote the event described by Lemma C.2. The following lemma provides a high-probability bound of estimation error terms.

**Lemma C.3.** On the event $\mathcal{E}_{C.2}$, we have for any $k \in [K], h \in [H]$,

$$Q_{k,h}(s_h^k, a_h^k) - r(s_h^k, a_h^k) - \mathbb{P}V_{k,h+1}(s_h^k, a_h^k) \leq 2 \min\left\{1, \widehat{\beta}_k \|\widehat{\boldsymbol{\Sigma}}_{k,0}^{-1/2} \boldsymbol{\phi}_{k,h,0}\|_2\right\}.$$

*Proof of Lemma C.3.* The proof is almost the same as that of Lemma C.4 in Zhou & Gu (2022), only to replace $V_{k,h}(s_h^k)$ with $Q_{k,h}(s_h^k, a_h^k)$. □

## C.2 RECURSIVE BOUNDS FOR STOCHASTIC POLICY

For any $k \in [k], h \in [H]$, we define the indicator function $I_h^k$ as the following

$$I_h^k := \mathbb{1}\left\{\forall m \in \overline{[M]}, \det(\widehat{\boldsymbol{\Sigma}}_{k,m}^{-1/2})/\det(\widetilde{\boldsymbol{\Sigma}}_{k,h,m}^{-1/2}) \leq 4\right\},$$

where $I_h^k$ is obviously $\mathcal{G}_h^k$-measurable and monotonically decreasing with respect to $h$ for all $k \in [K]$. For all $m \in \overline{[M]}$, we also define the following quantities:

$$R_m = \sum_{k=1}^{K}\sum_{h=1}^{H} I_h^k \min\left\{1, \widehat{\beta}_k \|\widehat{\boldsymbol{\Sigma}}_{k,m}^{-1/2} \boldsymbol{\phi}_{k,h,m}\|_2\right\}, \tag{C.2}$$

$$A_m = \sum_{k=1}^{K}\sum_{h=1}^{H} I_h^k \left[\mathbb{P}V_{k,h+1}^{2^m}(s_h^k, a_h^k) - V_{k,h+1}^{2^m}(s_{h+1}^k) + \mathbb{J}_{h+1}^k Q_{k,h+1}^{2^m}(s_{h+1}^k) - Q_{k,h+1}^{2^m}(s_{h+1}^k, a_{h+1}^k)\right], \tag{C.3}$$

$$S_m = \sum_{k=1}^{K}\sum_{h=1}^{H} I_h^k \left\{\mathbb{V}V_{k,h+1}^{2^m}(s_h^k, a_h^k) + \mathbb{J}_{h+1}^k Q_{k,h+1}^{2^{m+1}}(s_{h+1}^k) - \left[\mathbb{J}_{h+1}^k Q_{k,h+1}^{2^m}(s_{h+1}^k)\right]^2\right\}, \tag{C.4}$$

$$G = \sum_{k=1}^{K}(1 - I_H^k). \tag{C.5}$$

Finally, for simplicity, we also define

$$\iota = \log(1 + KH/(d\lambda\xi^2)), \tag{C.6}$$

$$\zeta = 4\log(4\log(KH)/\delta). \tag{C.7}$$

**Remark C.4.** Our definition is nearly the same as in Zhou & Gu (2022), despite that in our algorithm we use stochastic policies, which induce an additional term each in $A_m$ and $S_m$, regarding to the random variable and its conditional variance of the policies noise.

Now we are going to bound all these quantities. Basically, the technique we are using is nearly the same as in Zhou & Gu (2022). The only difference is that we need to deal with the extra policy noise resulted by stochastic policy.

**Lemma C.5.** Let $\gamma, \xi$ be defined in Algorithm 1, then for $m \in \overline{[M-1]}$, we have

$$R_m \leq \min\{8d\iota + 8\widehat{\beta}_K \gamma^2 d\iota + 8\widehat{\beta}_K \sqrt{d\iota}\sqrt{S_m + 4R_m + 2R_{m+1} + KH\xi^2}, KH\}. \tag{C.8}$$

We also have $R_{M-1} \leq KH$.

*Proof.* For $(k, h)$ such that $I_h^k = 1$, using Lemma D.2, we have

$$\left\|\widehat{\mathbf{\Sigma}}_{k,m}^{-1/2} \boldsymbol{\phi}_{k,h,m}\right\|_2 \leq \left\|\widetilde{\mathbf{\Sigma}}_{k,k,m}^{-1/2} \boldsymbol{\phi}_{k,h,m}\right\|_2 \cdot \sqrt{\frac{\det(\widehat{\mathbf{\Sigma}}_{k,m}^{-1})}{\det(\widetilde{\mathbf{\Sigma}}_{k,m}^{-1})}} \leq 4\left\|\widetilde{\mathbf{\Sigma}}_{k,k,m}^{-1/2} \boldsymbol{\phi}_{k,h,m}\right\|_2.$$

Substituting the above inequality into (C.2), we have

$$R_m \leq 4\sum_{k=1}^{K}\sum_{h=1}^{H} \min\left\{1, I_h^k \widehat{\beta}_k \left\|\widetilde{\mathbf{\Sigma}}_{k,h,m}^{-1/2} \boldsymbol{\phi}_{k,h,m}\right\|_2\right\},$$

where the right hand side can be bounded by Lemma C.1, with $\beta_{k,h} = I_h^k \widehat{\beta}_k, \bar{\sigma}_{k,h} = \bar{\sigma}_{k,h,m}, \mathbf{x}_{k,h} = \boldsymbol{\phi}_{k,h,m}$ and $\mathbf{Z}_{k,h} = \widetilde{\mathbf{\Sigma}}_{k,h,m}$. We have

$$\sum_{k=1}^{K}\sum_{h=1}^{H} \min\left\{1, I_h^k \widehat{\beta}_k \left\|\widetilde{\mathbf{\Sigma}}_{k,h,m}^{-1/2} \boldsymbol{\phi}_{k,h,m}\right\|_2\right\}$$

$$\leq 2d\iota + 2\widehat{\beta}_K \gamma^2 d\iota + 2\widehat{\beta}_K \sqrt{d\iota}\sqrt{\sum_{k=1}^{K}\sum_{h=1}^{H} I_h^k [\bar{\mathbb{V}} V_{k,h+1}^{2^m}(s_h^k, a_h^k) + E_{k,h,m}] + KH\xi^2}$$

$$\leq 2d\iota + 2\widehat{\beta}_K \gamma^2 d\iota + 2\widehat{\beta}_K \sqrt{d\iota}\sqrt{\sum_{k=1}^{K}\sum_{h=1}^{H} I_h^k [\mathbb{V} V_{k,h+1}^{2^m}(s_h^k, a_h^k) + 2E_{k,h,m}] + KH\xi^2}$$

$$\leq 2d\iota + 2\widehat{\beta}_K \gamma^2 d\iota + 2\widehat{\beta}_K \sqrt{d\iota}\sqrt{\sum_{k=1}^{K}\sum_{h=1}^{H} I_h^k \mathbb{V} V_{k,h+1}^{2^m}(s_h^k, a_h^k) + 4R_m + 2R_{m+1} + KH\xi^2}. \tag{C.9}$$

Since we have

$$\sum_{k=1}^{K}\sum_{h=1}^{H} I_h^k \left[\mathbb{J}_{h+1}^k Q_{k,h+1}^{2^{m+1}}(s_{h+1}^k) - \left(\mathbb{J}_{h+1}^k Q_{k,h+1}^{2^m}(s_{h+1}^k)\right)^2\right] \geq 0,$$

which, substituted into (C.4), gives

$$\sum_{k=1}^{K}\sum_{h=1}^{H} I_h^k \mathbb{V} V_{k,h+1}^{2^m}(s_h^k, a_h^k) \leq S_m. \tag{C.10}$$

Therefore by substituting (C.10) into (C.9), we have

$$R_m \leq 8d\iota + 8\widehat{\beta}_K \gamma^2 d\iota + 8\widehat{\beta}_K \sqrt{d\iota}\sqrt{\sum_{k=1}^{K}\sum_{h=1}^{H} I_h^k \mathbb{V} V_{k,h+1}^{2^m}(s_h^k, a_h^k) + 4R_m + 2R_{m+1} + KH\xi^2}$$

$$\leq 8d\iota + 8\widehat{\beta}_K \gamma^2 d\iota + 8\widehat{\beta}_K \sqrt{d\iota}\sqrt{S_m + 4R_m + 2R_{m+1} + KH\xi^2},$$

which completes the proof. $\qquad \square$

**Lemma C.6.** On the event $\mathcal{E}_{C.2}$, for all $m \in \overline{[M-1]}$, we have

$$S_m \le |A_{m+1}| + 2^{m+1}(K + 2R_0) + G.$$

*Proof.* The proof follows the proof of Lemma C.6 in Zhou & Gu (2022) and Lemma 25 in Zhang et al. (2021b). We have

$$
S_m = \sum_{k=1}^{K} \sum_{h=1}^{H} I_h^k \left\{ \mathbb{V} V_{k,h+1}^{2^m}(s_h^k, a_h^k) + \mathbb{J}_{h+1}^k Q_{k,h+1}^{2^{m+1}}(s_{h+1}^k) - \left[ \mathbb{J}_{h+1}^k Q_{k,h+1}^{2^m}(s_{h+1}^k) \right]^2 \right\}
$$

$$
= \sum_{k=1}^{K} \sum_{h=1}^{H} I_h^k \left( \mathbb{P} V_{k,h+1}^{2^{m+1}}(s_h^k, a_h^k) - Q_{k,h+1}^{2^{m+1}}(s_{h+1}^k, a_{h+1}^k) \right)
$$

$$
+ \sum_{k=1}^{K} \sum_{h=1}^{H} I_h^k \left[ Q_{k,h}^{2^{m+1}}(s_h^k, a_h^k) - \left( [\mathbb{P} V_{k,h+1}^{2^m}](s_h^k, a_h^k) \right)^2 \right]
$$

$$
+ \sum_{k=1}^{K} \sum_{h=1}^{H} I_h^k \left[ Q_{k,h+1}^{2^{m+1}}(s_{h+1}^k, a_{h+1}^k) - Q_{k,h}^{2^{m+1}}(s_h^k, a_h^k) \right]
$$

$$
+ \sum_{k=1}^{K} \sum_{h=1}^{H} I_h^k \left\{ \mathbb{J}_{h+1}^k Q_{k,h+1}^{2^{m+1}}(s_{h+1}^k) - \left[ \mathbb{J}_{h+1}^k Q_{k,h+1}^{2^m}(s_{h+1}^k) \right]^2 \right\}
$$

$$
= \underbrace{\sum_{k=1}^{K} \sum_{h=1}^{H} I_h^k \left[ \mathbb{P} V_{k,h+1}^{2^{m+1}}(s_h^k, a_h^k) - V_{k,h+1}^{2^{m+1}}(s_{h+1}^k) + \mathbb{J}_{h+1}^k Q_{k,h+1}^{2^{m+1}}(s_{h+1}^k) - Q_{k,h+1}^{2^{m+1}}(s_{h+1}^k) \right]}_{A_{m+1}}
$$

$$
+ \sum_{k=1}^{K} \sum_{h=1}^{H} I_h^k \left[ Q_{k,h}^{2^{m+1}}(s_h^k, a_h^k) - \left( [\mathbb{P} V_{k,h+1}^{2^m}](s_h^k, a_h^k) \right)^2 \right]
$$

$$
+ \sum_{k=1}^{K} \sum_{h=1}^{H} I_h^k \left[ Q_{k,h+1}^{2^{m+1}}(s_{h+1}^k, a_{h+1}^k) - Q_{k,h}^{2^{m+1}}(s_h^k, a_h^k) \right]
$$

$$
+ \sum_{k=1}^{K} \sum_{h=1}^{H} I_h^k \left( V_{k,h+1}^{2^{m+1}}(s_{h+1}^k) - [\mathbb{J}_{h+1}^k Q_{k,h+1}^{2^m}(s_{h+1}^k)]^2 \right).
$$

The first term here is exactly $A_{m+1}$, so we have

$$
S_m = A_{m+1} + \sum_{k=1}^{K} \sum_{h=1}^{H} I_h^k \left[ Q_{k,h}^{2^{m+1}}(s_h^k, a_h^k) - \left( [\mathbb{P} V_{k,h+1}^{2^m}](s_h^k, a_h^k) \right)^2 \right]
$$

$$
+ \sum_{k=1}^{K} \sum_{h=1}^{H} I_h^k \left[ Q_{k,h+1}^{2^{m+1}}(s_{h+1}^k, a_{h+1}^k) - Q_{k,h}^{2^{m+1}}(s_h^k, a_h^k) \right]
$$

$$
+ \sum_{k=1}^{K} \sum_{h=1}^{H} I_h^k \left( V_{k,h+1}^{2^{m+1}}(s_{h+1}^k) - [\mathbb{J}_{h+1}^k Q_{k,h+1}^{2^m}(s_{h+1}^k)]^2 \right)
$$

$$
\le A_{m+1} + \sum_{k=1}^{K} \sum_{h=1}^{H} I_h^k \left[ Q_{k,h}^{2^{m+1}}(s_h^k, a_h^k) - \left( [\mathbb{P} V_{k,h+1}^{2^m}](s_h^k, a_h^k) \right)^2 \right]
$$

$$
+ \sum_{k=1}^{K} I_{h_k}^k Q_{k,h_k+1}^{2^{m+1}}(s_{h_k+1}^k, a_{h_k+1}^k) + \sum_{k=1}^{K} \sum_{h=1}^{H} I_h^k \left\{ V_{k,h+1}^{2^{m+1}}(s_{h+1}^k) - [\mathbb{J}_{h+1}^k Q_{k,h+1}^{2^m}(s_{h+1}^k)]^2 \right\},
$$

where $h_k$ is the largest index satisfying $I_h^k = 1$. If $h_k < H$, we have $I_{h_k}^k Q_{k,h_k+1}^{2^{m+1}}(s_{h_k+1}^k, a_{h_k+1}^k) \le 1 = 1 - I_H^k$ and if $h_k = H$, we have $I_{h_k}^k Q_{k,h_k+1}^{2^{m+1}}(s_{h_k+1}^k, a_{h_k+1}^k) = 0 = 1 - I_H^k$, so in both

circumstances we have

$$S_m \le A_m + \underbrace{\sum_{k=1}^{K}\sum_{h=1}^{H} I_h^k [Q_{k,h}^{2^{m+1}}(s_h^k, a_h^k) - ([\mathbb{P}V_{k,h+1}^{2^m}](s_h^k, a_h^k))^2]}_{(ii)} + \underbrace{\sum_{k=1}^{K}(1 - I_H^k)}_{G}$$

$$+ \underbrace{\sum_{k=1}^{K}\sum_{h=1}^{H} I_h^k [V_{k,h+1}^{2^{m+1}}(s_{h+1}^k) - [\mathbb{J}_{h+1}^k Q_{k,h+1}^{2^m}(s_{h+1}^k)]^2]}_{(iv)}. \tag{C.11}$$

For (ii) in (C.11), we have

$$\sum_{k=1}^{K}\sum_{h=1}^{H} I_h^k \left[ Q_{k,h}^{2^{m+1}}(s_h^k, a_h^k) - \left([\mathbb{P}V_{k,h+1}^{2^m}](s_h^k, a_h^k)\right)^2 \right]$$

$$\le \sum_{k=1}^{K}\sum_{h=1}^{H} I_h^k \left[ Q_{k,h}^{2^{m+1}}(s_h^k, a_h^k) - \left([\mathbb{P}V_{k,h+1}](s_h^k, a_h^k)\right)^{2^{m+1}} \right]$$

$$= \sum_{k=1}^{K}\sum_{h=1}^{H} I_h^k (Q_{k,h}(s_h^k, a_h^k) - [\mathbb{P}V_{k,h+1}](s_h^k, a_h^k)) \prod_{i=0}^{m} (Q_{k,h}^{2^i}(s_h^k, a_h^k) + ([\mathbb{P}V_{k,h+1}](s_h^k, a_h^k))^{2^i})$$

$$\le 2^{m+1} \sum_{k=1}^{K}\sum_{h=1}^{H} I_h^k (r^k(s_h^k, a_h^k) + 2\min\{1, \widehat{\beta}_k \|\widehat{\Sigma}_{k,m}^{-1/2} \phi_{k,h,0}\|_2\})$$

$$\le 2^{m+1}(K + 2R_0),$$

where the first inequality holds by recursively using $\mathbb{E}X^2 \ge (\mathbb{E}^2 X)$, the second holds due to Assumption 3.1 and the third holds due to Lemma C.3. It remains to bound the last term (iv) in (C.11). We have

$$\sum_{k=1}^{K}\sum_{h=1}^{H} I_h^k [V_{k,h+1}^{2^{m+1}}(s_{h+1}^k) - [\mathbb{J}_{h+1}^k Q_{k,h+1}^{2^m}(s_{h+1}^k)]^2]$$

$$= \sum_{k=1}^{K}\sum_{h=1}^{H} I_h^k [(\mathbb{J}_{h+1}^k Q_{k,h+1}(s_{h+1}^k))^{2^{m+1}} - [\mathbb{J}_{h+1}^k Q_{k,h+1}^{2^m}(s_{h+1}^k)]^2]$$

$$= \sum_{k=1}^{K}\sum_{h=1}^{H} ((\mathbb{E}_{a \sim \pi_{h+1}^k(\cdot|\mathcal{G}_{k,h+1})}[I_h^k Q_{k,h+1}(s_{h+1}^k, a)|\mathcal{G}_{k,h+1}])^{2^{m+1}}$$

$$- \mathbb{E}_{a \sim \pi_{h+1}^k(\cdot|s_{h+1}^k)}^2 [(I_h^k Q_{k,h+1}(s_{h+1}^k, a))^{2^m}|s_{h+1}^k])$$

$$\le 0,$$

where the first equality holds due to the definition of $V_{k,h+1}(s_{h+1}^k)$, the second holds due to $I_h^k$ is $\mathcal{G}_{k,h+1}$-measurable, and the inequality holds due to $\mathbb{E}X^2 \ge (\mathbb{E}^2 X)$. Combining the estimations of the four terms completes the proof. $\square$

**Lemma C.7** (Lemma 25, Zhang et al. 2021b). We have $\mathbb{P}(\mathcal{E}_{C.7}) > 1 - 2M\delta$, where

$$\mathcal{E}_{C.7} := \{\forall m \in \overline{[M]}, |A_m| \le \min\{\sqrt{2\zeta S_m} + \zeta, 2KH\}\}.$$

*Proof.* The proof follows the proof of Lemma 25 in Zhang et al. (2021b). First, we define

$$\begin{cases} x_{k,h} := I_h^k \left[ \mathbb{P}V_{k,h+1}^{2^m}(s_h^k, a_h^k) - V_{k,h+1}^{2^m}(s_{h+1}^k) \right], \\ y_{k,h} := I_h^k \left[ \mathbb{J}_{h+1}^k Q_{k,h+1}^{2^m}(s_{h+1}^k) - Q_{k,h+1}^{2^m}(s_{h+1}^k, a_{h+1}^k) \right] \end{cases} \implies \begin{cases} \mathbb{E}[x_{k,h}|\mathcal{F}_{k,h}] = 0, \\ \mathbb{E}[y_{k,h}|\mathcal{G}_{k,h+1}] = 0. \end{cases}$$

Obviously, $\{x_{1,1}, y_{1,1}, ..., x_{k,h}, y_{k,h}\}$ forms a martingale difference sequence, whose conditional second-order moments are

$$\mathbb{E}[x_{k,h}^2|\mathcal{F}_{k,h}] = I_h^k[\mathbb{V}V_{k,h+1}^{2^m}(s_h^k, a_h^k)],$$

$$\mathbb{E}[y_{k,h}^2|\mathcal{G}_{k,h+1}] = I_h^k[\mathbb{J}_{h+1}^k Q_{k,h+1}^{2^{m+1}}(s_{h+1}^k) - (\mathbb{J}_{h+1}^k Q_{k,h+1}^{2^m}(s_{h+1}^k))^2].$$

Summing these terms over $[K] \times [H]$ yields

$$\sum_{k=1}^{K} \sum_{h=1}^{H} (\mathbb{E}[x_{k,h}^2 | \mathcal{F}_{k,h}] + \mathbb{E}[y_{k,h}^2 | \mathcal{G}_{k,h+1}]) = S_m. \qquad (C.12)$$

Therefore, by Lemma D.3, for each $m \in \overline{[M]}$, with probability at least $1 - \delta$, we have

$$A_m \le \sqrt{2\zeta S_m} + \zeta.$$

Taking union bound over $m \in \overline{[M]}$, and also using the fact that $|x_{k,h}|, |y_{k,h}| \le 1$ completes the proof. $\qquad \square$

**Lemma C.8** (Lemma C.8, Zhou & Gu 2022)**.** Let $G$ be defined in (C.5), then we have $G \le Md\iota/2$.

Finally wer provide the high-probability bounds of two remained martingales, both of which are direct application of Lemma D.1.

**Lemma C.9.** With probability at least $1 - \delta$, we have

$$\sum_{k=1}^{K} \left( \sum_{h=1}^{H} (r(s_h^k, a_h^k) - V_1^{\pi^k}(s_1^k) \right) \le \sqrt{2K \log(1/\delta)}.$$

**Lemma C.10.** With probability at least $1 - \delta$, we have

$$\sum_{k=1}^{K} \left( \mathbb{J}_1^k Q_{k,1}(s_1^k) - Q_{k,1}(s_1^k, a_1^k) \right) \le \sqrt{2K \log(1/\delta)}.$$

We use $\mathcal{E}_{C.9}$ and $\mathcal{E}_{C.10}$ to denote the event described by the corresponding lemmas.

C.3  PROOF OF THEOREM 5.1

Now we can proof our main result. First we are going to provide two theorems. The first theorem provides a horizon-free regret analysis of high-order moment estimator.

**Theorem C.11.** Set $M = \log(4KH)/\log 2$, for any $\delta > 0$, on event $\mathcal{E}_{C.2} \cap \mathcal{E}_{C.7} \cap \mathcal{E}_{C.9} \cap \mathcal{E}_{C.10}$, we have

$$\begin{aligned}
\text{Regret}(K) &\le 2432 \max\{32\widehat{\beta}_K^2 d\iota, \zeta\} + 192(d\iota + \widehat{\beta}_K \gamma^2 d\iota + \widehat{\beta}_K \sqrt{d\iota} \sqrt{Md\iota/2 + KH\alpha^2}) \\
&\quad + Md\iota/2 + 24(\sqrt{\zeta Md\iota} + \zeta) + [2\sqrt{2\log(1/\delta)} + 32\max\{8\widehat{\beta}_K \sqrt{d\iota}, \sqrt{2\zeta}\}]\sqrt{2K},
\end{aligned}$$

where $\iota, \zeta$ are defined in (C.6) and (C.7). Moreover, setting $\xi = \sqrt{d/(KH)}$, $\gamma = 1/d^{1/4}$ and $\lambda = d/B^2$ yields a bound $I_2 = \widetilde{O}(d\sqrt{K} + d^2)$ with high probability.

*Proof.* All the following proofs are under the event $\mathcal{E}_{C.2} \cap \mathcal{E}_{C.7} \cap \mathcal{E}_{C.9} \cap \mathcal{E}_{C.10}$. First, we have the composition for $I_2$, for all $k$, we define $Q_{k,H+1}(s,a) = 0$.

$$\sum_{k=1}^{K} V_{k,1}(s_1^k) = \sum_{k=1}^{K} \left( \mathbb{J}_1^k Q_{k,1}(s_1^k) - Q_{k,1}(s_1^k, a_1^k) \right)$$

$$+ \sum_{k=1}^{K} \sum_{h=1}^{H} \left( Q_{k,h}(s_h^k, a_h^k) - Q_{k+1,h+1}(s_{h+1}^k, a_{h+1}^k) \right)$$

$$= \sum_{k=1}^{K} \left( \mathbb{J}_1^k Q_{k,1}(s_1^k) - Q_{k,1}(s_1^k, a_1^k) \right)$$

$$+ \sum_{k=1}^{K} \sum_{h=1}^{H} I_h^k \left( Q_{k,h}(s_h^k, a_h^k) - Q_{k+1,h+1}(s_{h+1}^k, a_{h+1}^k) \right) \tag{C.13}$$

$$+ \sum_{k=1}^{K} \sum_{h=1}^{H} (1 - I_h^k) \left( Q_{k,h}(s_h^k, a_h^k) - Q_{k+1,h+1}(s_{h+1}^k, a_{h+1}^k) \right)$$

$$\leq \sum_{k=1}^{K} \left( \mathbb{J}_1^k Q_{k,1}(s_1^k) - Q_{k,1}(s_1^k, a_1^k) \right) + \sum_{k=1}^{K} (1 - I_{h_k}^k) Q_{k,h_k}(s_{h_k}^k, a_{h_k}^k)$$

$$+ \sum_{k=1}^{K} \sum_{h=1}^{H} I_h^k \left( Q_{k,h}(s_h^k, a_h^k) - Q_{k+1,h+1}(s_{h+1}^k, a_{h+1}^k) \right),$$

where $h_k$ is the smallest number such that $I_{h_k}^k = 0$. Then for the second term we have

$$\sum_{k=1}^{K} \sum_{h=1}^{H} I_h^k \left( Q_{k,h}(s_h^k, a_h^k) - Q_{k+1,h+1}(s_{h+1}^k, a_{h+1}^k) \right)$$

$$= \sum_{k=1}^{K} \sum_{h=1}^{H} I_h^k [r(s_h^k, a_h^k)] + \sum_{k=1}^{K} \sum_{h=1}^{H} I_h^k [Q_{k,h}(s_h^k, a_h^k) - r(s_h^k, a_h^k) - \mathbb{P}V_{k,h+1}(s_h^k, a_h^k)]$$

$$+ \sum_{k=1}^{K} \sum_{h=1}^{H} I_h^k [\mathbb{P}V_{k,h+1}(s_h^k, a_h^k) - V_{k,h+1}(s_{h+1}^k) + \mathbb{J}_{h+1}^k Q(s_{h+1}^k) - Q_{k,h+1}(s_{h+1}^k, a_{h+1}^k)]$$

$$\leq \sum_{k=1}^{K} \sum_{h=1}^{H} r(s_h^k, a_h^k) + \sum_{k=1}^{K} \sum_{h=1}^{H} I_h^k [Q_{k,h}(s_h^k, a_h^k) - r(s_h^k, a_h^k) - \mathbb{P}V_{k,h+1}(s_h^k, a_h^k)] + A_0.$$

Substituting the inequality above to (C.13), we have:

$$\sum_{k=1}^{K} \left( V_{k,1}(s_1^k) - V_1^{\pi^k}(s_1^k) \right)$$

$$\leq \sum_{k=1}^{K} \left( \mathbb{J}_1^k Q_{k,1}(s_1^k) - Q_{k,1}(s_1^k, a_1^k) \right) + \sum_{k=1}^{K} (1 - I_H^k) + \sum_{k=1}^{K} \left( \sum_{h=1}^{H} (r(s_h^k, a_h^k) - V_1^{\pi^k}(s_1^k) \right)$$

$$+ \sum_{k=1}^{K} \sum_{h=1}^{H} I_h^k [Q_{k,h}(s_h^k, a_h^k) - r(s_h^k, a_h^k) - \mathbb{P}V_{k,h+1}(s_h^k, a_h^k)] + A_0$$

$$\leq 2\sqrt{2K \log(1/\delta)} + G + 2R_0 + A_0,$$

where the second inequality holds due to Lemma C.10, Lemma C.9 and Lemma C.3.

Thus, we only need to bound $2R_0 + A_0$. We have

$$|A_m| \leq \sqrt{2\zeta S_m} + \zeta$$

$$\leq \sqrt{2\zeta(|A_{m+1}| + G + 2^{m+1}(K + 2R_0)} + \zeta$$

$$\leq \sqrt{2\zeta} \sqrt{|A_{m+1}| + 2^{m+1}(K + 2R_0)} + \sqrt{2\zeta G} + \zeta,$$

where the first inequality holds due to Lemma C.7, the second inequality holds due to Lemma C.6 and the third holds due to $\sqrt{a+b} \leq \sqrt{a} + \sqrt{b}$. We also have:

$$
\begin{aligned}
R_m &\leq 8d\iota + 8\widehat{\beta}_K\gamma^2 d\iota + 8\widehat{\beta}_K\sqrt{d\iota}\sqrt{S_m + 4R_m + 2R_{m+1} + KH\alpha^2} \\
&\leq 8\widehat{\beta}_K\sqrt{d\iota}\sqrt{|A_{m+1}| + G + 2^{m+1}(K + 2R_0) + 4R_m + 2R_{m+1} + KH\alpha^2} \\
&\quad + 8d\iota + 8\widehat{\beta}_K\gamma^2 d\iota \\
&\leq 8\widehat{\beta}_K\sqrt{d\iota}\sqrt{|A_{m+1}| + 2^{m+1}(K + 2R_0) + 4R_m + 2R_{m+1}} \\
&\quad + 8d\iota + 8\widehat{\beta}_K\gamma^2 d\iota + 8\widehat{\beta}_K\sqrt{d\iota}\sqrt{G + KH\alpha^2},
\end{aligned}
$$

where the first inequality holds due to Lemma C.5, the second holds due to Lemma C.6 and we denote $I_c = 8d\iota + 8\widehat{\beta}_K\gamma^2 d\iota + 8\widehat{\beta}_K\sqrt{d\iota}\sqrt{G + KH\alpha^2} + \sqrt{2\zeta G} + \zeta$. Combining the two estimations we have

$$
\begin{aligned}
|A_m| + 2R_m &\leq 2I_c + \sqrt{2}\max\{8\widehat{\beta}_K\sqrt{d\iota}, \sqrt{2\zeta}\} \\
&\quad \sqrt{5|A_{m+1}| + 5 \cdot 2^{m+1}(K + 2R_0) + 16R_m + 8R_{m+1}} \\
&\leq 2I_c + 4\max\{8\widehat{\beta}_K\sqrt{d\iota}, \sqrt{2\zeta}\} \\
&\quad \sqrt{|A_{m+1}| + 2R_{m+1} + |A_m| + 2R_m + 2^{m+1}(K + 2R_0 + |A_0|)},
\end{aligned}
$$

where the first inequality holds due to $\sqrt{a} + \sqrt{b} \leq \sqrt{2(a+b)}$. Then by Lemma D.4, with $a_m = 2|A_m| + R_m \leq 4KH$ and $M = \log(4KH)/\log 2$, we have:

$$
\begin{aligned}
|A_0| + 2R_0 &\leq 22 \cdot 16\max\{64\widehat{\beta}_K^2 d\iota, 2\zeta\} + 12I_c \\
&\quad + 16\max\{8\widehat{\beta}_K\sqrt{d\iota}, \sqrt{2\zeta}\}\sqrt{2(K + 2R_0 + |A_0|)} \\
&\leq 704\max\{32\widehat{\beta}_K^2 d\iota, \zeta\} \\
&\quad + 12(8d\iota + 8\widehat{\beta}_K\gamma^2 d\iota + 8\widehat{\beta}_K\sqrt{d\iota}\sqrt{G + KH\alpha^2} + \sqrt{2\zeta G} + \zeta) \\
&\quad + 16\max\{8\widehat{\beta}_K\sqrt{d\iota}, \sqrt{2\zeta}\}\sqrt{2K} + 16\sqrt{2}\max\{8\widehat{\beta}_K\sqrt{d\iota}, \sqrt{2\zeta}\}\sqrt{2R_0 + |A_0|}.
\end{aligned}
$$

By the fact that $x \leq a\sqrt{x} + b \Rightarrow x \leq 2a^2 + 2b$, we have

$$
\begin{aligned}
|A_0| + 2R_0 &\leq 2432\max\{32\widehat{\beta}_K^2 d\iota, \zeta\} \\
&\quad + 24(8d\iota + 8\widehat{\beta}_K\gamma^2 d\iota + 8\widehat{\beta}_K\sqrt{d\iota}\sqrt{G + KH\alpha^2} + \sqrt{2\zeta G} + \zeta) \\
&\quad + 32\max\{8\widehat{\beta}_K\sqrt{d\iota}, \sqrt{2\zeta}\}\sqrt{2K}.
\end{aligned}
$$

Bounding $G$ by Lemma C.8, we have

$$
\begin{aligned}
\sum_{k=1}^{K}\left(V_{k,1}(s_1^k) - V_1^{\pi^k}(s_1^k)\right) &\leq 2\sqrt{2K\log(1/\delta)} + G + 2R_0 + A_0 \\
&\leq 2432\max\{32\widehat{\beta}_K^2 d\iota, \zeta\} + 192(d\iota + \widehat{\beta}_K\gamma^2 d\iota + \widehat{\beta}_K\sqrt{d\iota}\sqrt{Md\iota/2 + KH\alpha^2}) \\
&\quad + Md\iota/2 + 24(\sqrt{\zeta Md\iota} + \zeta) + [2\sqrt{2\log(1/\delta)} + 32\max\{8\widehat{\beta}_K\sqrt{d\iota}, \sqrt{2\zeta}\}]\sqrt{2K},
\end{aligned}
$$

which completes the proof. $\qquad\square$

**Theorem C.12.** On the event $\mathcal{E}_{C.2}$, we have

$$
\sum_{k=1}^{K}(V_{k,1}^*(s_1) - \bar{V}_{k,1}(s_1)) \leq \frac{H\log|\mathcal{S}|^2|\mathcal{A}|}{\alpha} + \frac{K\alpha}{2H}.
$$

*Proof.* This follows the standard regret analysis of online mirror descent. The only difference from standard arguments is that we need to deal with the changing convex set. We include the adapted proof for completeness. For sake of brevity, we denote $f_k(z) = \sum_{h,s,a,s'} z_h(s,a,s')r^k(s,a)$, then we have

$$
f_k(z^*) = V_{k,1}^*(s_1), \quad f_k(z^k) = \bar{V}_{k,1}(s_1), \quad \nabla f_k(\cdot) = (r_h^k(s,a))_{s,a,s',h},
$$

where $z^*$ is the occupancy measure induced by $\pi^*$ and true transition. Since we have that for all $k \in [1:K]$, $\theta^* \in \mathcal{C}_k$, we know that $z^* \in D_k$ for all $k$. Then we have

$$
\begin{aligned}
f_k(z^*) - f_k(z^k) &= \nabla f_k(z^k)^\top (z^* - z^k) \\
&= \alpha^{-1}(\nabla \Phi(w^{k+1}) - \nabla \Phi(z^k))^\top (z^k - z^*) \\
&= \alpha^{-1}(D_\Phi(z^*||z^k) + D_\Phi(z^k||w^{k+1}) - D_\Phi(x^*||w^{k+1})),
\end{aligned}
$$

where the equities hold due to the update rule of mirror descent. Because $D_{k+1}$ is convex and $z^* \in D_{k+1}$, we have the first order optimality for $z^{k+1}$:

$$
(\nabla \Phi(z^{k+1}) - \nabla \Phi(w^{k+1}))^\top (z^{k+1} - z^*) \leq 0,
$$

which can be written equivalently as the generalized Pythagorean inequality:

$$
D_\Phi(z^*||w^{k+1}) \geq D_\Phi(z^*||z^{k+1}) + D_\Phi(z^{k+1}||w^{k+1}).. \tag{C.14}
$$

Combining the two expression, we have

$$
f_k(z^*) - f_k(z^k) \leq \alpha^{-1}(D_\Phi(z^*||z^k) - D_\Phi(z^*||z^{k+1})) + \alpha^{-1}(D_\Phi(z^k||w^{k+1}) - D_\Phi(z^{k+1}||w^{k+1})).
$$

For the second term, we have

$$
\begin{aligned}
&D_\Phi(z^k||w^{k+1}) - D_\Phi(z^{k+1}||w^{k+1}) \\
&= \Phi(z^k) - \Phi(z^{k+1}) - \nabla \Phi(w^{k+1})^\top (z^k - z^{k+1}) \\
&\leq (\nabla \Phi(z^k) - \nabla \Phi(w^{k+1})^\top (z^k - z^{k+1}) - \frac{1}{2H}\left\|z^k - z^{k+1}\right\|_1^2 \\
&= \alpha \nabla f_k^\top (z^k - z^{k+1}) - \frac{1}{2H}\left\|z^k - z^{k+1}\right\|_1^2 \\
&\leq \frac{\alpha}{H}\left\|z^k - z^{k+1}\right\|_1 - \frac{1}{2H}\left\|z^k - z^{k+1}\right\|_1^2 \\
&\leq \frac{\alpha^2}{2H},
\end{aligned}
$$

where the first inequality holds due to Lemma 4.2, the second inequality holds due to $r^k(\cdot,\cdot) \leq 1/H$, and the third inequality holds due to quadratic inequality.

Summing up over $k$, we have

$$
\begin{aligned}
\sum_{k=1}^{K}(f_k(z^*) - f_k(z^k)) &\leq \alpha^{-1}(D_\Phi(z^*||z^1) - D_\Phi(z^*||z^{K+1})) + \frac{\alpha K}{2H} \\
&\leq \frac{D_\Phi(z^*||z^1)}{\alpha} + \frac{K\alpha}{2H} \\
&\leq \frac{D_\Phi(z^*||w^1)}{\alpha} + \frac{K\alpha}{2H} \\
&\leq \frac{H \log |S|^2 |A|}{\alpha} + \frac{K\alpha}{2H},
\end{aligned}
$$

where the third inequality holds due to extended Pythagorean's inequality (C.14) and the forth holds since $w_h^1 = z_h^0$ is an uniform distribution on $\mathcal{S} \times \mathcal{A} \times \mathcal{S}$. $\qquad\square$

Now we are able to prove our main result.

*Proof of Theorem 5.1.* First we have the following regret decomposition

$$
\begin{aligned}
\sum_{k=1}^{K}\left(V_{k,1}^*(s_1) - V_1^{\pi_k}(s_1)\right) &= \sum_{k=1}^{K}\left(V_{k,1}^*(s_1) - \bar{V}_{k,1}(s_1) + \bar{V}_{k,1}(s_1) - V_{k,1}(s_1) + V_{k,1}(s_1) - V_1^{\pi_k}(s_1)\right) \\
&\leq \underbrace{\sum_{k=1}^{K}\left(V_{k,1}^*(s_1) - \bar{V}_{k,1}(s_1)\right)}_{I_1} + \underbrace{\sum_{k=1}^{K}\left(V_{k,1}(s_1) - V_1^{\pi_k}(s_1)\right)}_{I_2},
\end{aligned}
$$

where the inequality holds due to Lemma 6.1. Picking $\xi = \sqrt{d/(KH)}, \gamma = 1/d^{1/4}$ and $\lambda = d/B^2$, by Theorem C.11, we know that $I_2 = \widetilde{O}(d\sqrt{K} + d^2)$ on event $\mathcal{E}_{C.2} \cap \mathcal{E}_{C.7} \cap \mathcal{E}_{C.9} \cap \mathcal{E}_{C.10}$. By Theorem C.12, we have

$$I_1 \leq \frac{H \log |\mathcal{S}|^2 |\mathcal{A}|}{\alpha} + \frac{K\alpha}{2H}.$$

Setting $\alpha = H/\sqrt{K}$, combining the two terms and taking the union bound of event $\mathcal{E}_{C.2} \cap \mathcal{E}_{C.7} \cap \mathcal{E}_{C.9} \cap \mathcal{E}_{C.10}$ completes the proof. $\qquad\square$

### C.4 PROOF OF THEOREMS 5.3

*Proof of Theorem 5.3.* The major idea is to cast learning a special MDP with finite $\mathcal{S}, \mathcal{A}$ and deterministic (and known) transition, which can be represented as a *complete $|\mathcal{A}|$-way tree*, as *prediction with expert advice* and leverage the asymptotic lower bound (Cesa-Bianchi & Lugosi, 2006, Theorem 3.7) to manifest a $\sqrt{HK \log |\mathcal{A}|}$ or $\sqrt{K \log |\mathcal{S}|}$ dependence in the lower bound. Our two-stage reduction begins with a hard-to-learn MDP $M_1$ with its total reward in each episode bounded by 1. The hard instance $M_1(\mathcal{S}, \mathcal{A}, H, \{r_h^k\}, \mathbb{P})$ is purely deterministic, where $H$ is even, i.e., $\forall a \in \mathcal{A}, s, s' \in \mathbb{P}(s'|s, a)$ is either 0 or 1. The transition dynamics forms a complete $|\mathcal{A}|$-way tree with each node corresponding to a state and each edge directed to leaves corresponding to the transition after an action. Let $\mathcal{S}[l, m]$ denote the $m$-th state (node) in the $l$-th layer of the tree, $\forall l \in [H + 1], m \in [|\mathcal{A}|^l]$ and let $\mathcal{A}[l, m, n]$ denote the *only* action (edge) from $\mathcal{S}[l, m]$ to $\mathcal{S}[l + 1, (m - 1)|\mathcal{A}| + n], \forall l \in [H], m \in [|\mathcal{A}|^l], n \in [|\mathcal{A}|]$. The agent is forced to start from $s_1^k := \mathcal{S}[1, 1]$ in every episode $k \in [K]$ so it will always end up in a leaf state, which is denoted by $s_{H+1}^k := \mathcal{S}[H + 1, m_0]$ for some $m_0 \in [|\mathcal{A}|^H]$. To align with *prediction with expert advice*, we constrain $r_h^k(\cdot, \cdot) := 0, \forall h \in [H - 1]$ and $r_H^k(\cdot, \cdot) \in [0, 1]$, which implies the agent can not receive any positive reward until it is moving towards the last layer of the MDP (tree). Under these constraints, We allow $r^k$ to change arbitrarily across episodes.[4] Notice that unlike the common reward design in the hard instance constructions for obtaining information-theoretic lower bounds, which are usually to illustrate the difficulty of parameter estimation, we do not assign specific numeric values to $r_h^k$ in order to expose the impact of the adversarial environment.

All the $|\mathcal{A}|^H$ rewards towards leaves in $M_1$, $r_H^k(\cdot, \cdot)$, form an array of experts and any given policy $\pi^k = \{\pi_h^k(\cdot|\cdot)_{h=1}^H\}$ actually induces a probability simplex (of state-reaching after taking the action $a_{H-1}^k$) over these experts in episode $k$, which can be represented by a weight vector $w_k \in \Delta\left([|\mathcal{A}|^H]\right)$. Clearly, $V_{k,1}^{\pi^k}(s_1^k) = \langle w_k, r_H^k \rangle$, where we abuse $r_H^k$ to denote the reward vector $r_H^k \in [0, 1]^{|\mathcal{A}|^H}$ towards leaves corresponding to $w_k$. With hindsight, $\pi^* = \sup_\pi \sum_{k=1}^K V_{k,1}^\pi(s_1^k)$, by which the optimal weight vector $w_*$ is induced. In such a deterministic MDP, $\pi^*$ may not be unique but the corresponding $w_*$ can have a restricted support set over the $|\mathcal{A}|^H$ experts, which we re-index as $r_H^k[i]$. To be more rigorous, let $\mathbb{W} = \mathrm{supp}\, w_* := \{i \in [|\mathcal{A}|^H] : w_*[i] \neq 0\}$, then obviously $\mathbb{W} = \mathrm{argmax}_i \sum_{k=1}^K r_H^k[i]$. Thus, $\forall i \in \mathbb{W}, \sum_{k=1}^K V_{k,1}^*(s_1^k) = \sum_{k=1}^K \langle w_*, r_H^k \rangle = \sum_{k=1}^K r_H^k[i] = \max_j \sum_{k=1}^K r_H^k[j]$ and

$$\mathrm{Regret}(K) := \sum_{k=1}^K V_{k,1}^*(s_1^k) - V_{k,1}^{\pi^k}(s_1^k) = \max_{i \in [|\mathcal{A}|^H]} \sum_{k=1}^K r_H^k[i] - \langle w_k, r_H^k \rangle. \qquad (C.15)$$

(C.15) reveals the connection between learning in $M_1$ with its $|\mathcal{S}| = \Theta(|\mathcal{A}|^H)$ and *prediction with expert advice* with $|\mathcal{A}|^H$ experts and $K$ rounds. Each expert has its reward bounded in $[0, 1]$. The first stage of this reduction accounts for the overhead incurred by the adversary under full-information feedback. For any algorithm, there is a well-known asymptotic lower bound for $\mathrm{Regret}(K)$:

**Lemma C.13.** For any algorithm and any given nonempty action space $\mathcal{A}$, there exists an episodic MDP (with the corresponding $\mathcal{A}$) satisfying Assumption 3.2 such that its expected regret satisfies

$$\lim_{H\to\infty} \lim_{K\to\infty} \frac{\mathrm{Regret}(K)}{\sqrt{(HK/2) \log |\mathcal{A}|}} \geq 1,$$

---

[4]Here in the constructions of this proof, we allow the reward function to be time-inhomogeneous because although in Assumption 3.1 we set the reward to be time-homogeneous for the simplicity of notation, all the arguments in the proof of our regret upper bound can naturally be applicable to the time-inhomogenous case.

if the total reward in each episode is bounded in $[0, 1]$.

*Proof of Lemma C.13.* See the proof of Cesa-Bianchi & Lugosi (2006, Theorem 3.7) for details. The only work left is to verify Assumption 3.2. Let $d = 1, \theta = 1$ and the deterministic transition kernel $\mathbb{P}$ in $M_1$ be the only basic model in the linear mixture MDP, then we can see that the $M_1$ we construct indeed satisfies Assumption 3.2. □

We bridge the gap between the reward design in Lemma C.13 and Assumption 3.1 in Theorem 5.3 via the second stage of this reduction.

When $H$ is even, Lemma C.13 also holds for $\bar{M}_1 := M_1(\mathcal{S}, \mathcal{A}, H/2, \{\bar{r}_h^k\}, \mathbb{P})$ with $H$ replaced by $H/2$, where the $\mathcal{S}$, $\mathcal{A}$, and $\mathbb{P}$ from $M_1$ are restricted to the first $H/2$ time steps in $\bar{M}_1$ and $\bar{r}_{H/2}^k(\cdot, \cdot) \in [0, 1]$ and the agent gets no reward in all the first $H/2 - 1$ time steps by construction. We can equivalently transform $\bar{M}_1$ into a MDP $M_2$ satisfying Assumption 3.1 with planning horizon $H$ as follows. We replace every node $\mathcal{S}[H/2 + 1, \cdot]$ in the $(H/2 + 1)$-th layer of $\bar{M}_1$ by a $(H/2 + 1)$-layer complete $|\mathcal{A}|$-way tree, and further assign the transition kernel of $M_1$ to this extended $\bar{M}_1$. To obtain $M_2$, a refined reward design is to assign zero reward for actions (edges) conducted in states in the first $H/2$ layers and we assign each edge (action) in this subtree with a reward $\bar{r}_{H/2}^k (\mathcal{S}[H/2, m], \mathcal{A}[H/2, m, n])/H \in [0, 1/H]$ for any subtree rooted in $\mathcal{S}[H/2 + 1, (m-1)|\mathcal{A}| + n]$. Such a construction yields $M_2(\mathcal{S}, \mathcal{A}, H, \{\tilde{r}_h^k\}, \mathbb{P})$, learning in which can similarly be reduced to the standard *prediction with expert advice* with $|\mathcal{A}|^{H/2}$ experts and $K$ rounds. Therefore, Lemma C.13 also holds for $M_2$ with $H$ replaced by $H/2$, yet the properties of the reward assignment in $M_2$ is strictly strong than Assumption 3.1 in that all the actions conducted from states in the same subtree rooted in the $(H/2 + 1)$-th layer causes the same reward.

Our goal is to claim a lower bound for a $M_{3.1}(\mathcal{S}, \mathcal{A}, H, \{\hat{r}_h^k\}, \mathbb{P})$, which shares the same $\mathcal{S}$, $\mathcal{A}$, and $\mathbb{P}$ with $M_1$ but has its reward assignment generally satisfying Assumption 3.1, i.e. all actions taken from all states cause a reward $\hat{r}_h^k \in [0, 1/H]$. Since $M_2$ is strictly a special case of $M_{3.1}$, which implies that the asymptotic lower bound for $M_{3.1}$ can not be lower than that in Lemma C.13 up to a constant factor $\sqrt{2}$. Also, it is obvious that $|\mathcal{S}| = \Theta(|\mathcal{A}|^H)$ in a complete $|\mathcal{A}|$-way tree with $H + 1$ layers. □

## C.5 Proof of Theorem 5.5

*Proof of Theorem 5.5.* The proof is almost identical to the proof of Theorem 5.4 in Zhou & Gu (2022). Consider the MDP $M' = (\mathcal{S}, \mathcal{A}, H, r', \mathbb{P})$ constructed in Theorem 5.4, Zhou & Gu (2022). Now we consider a linear mixture MDP with adversarial reward $M' = (\mathcal{S}, \mathcal{A}, H, \{r_k\}_{k \in [K]}, \mathbb{P})$, where all the elements except reward function is inherited from $M'$. Now we define $r^k(\cdot, \cdot) = r'(\cdot, \cdot)$ for all $k \in [K]$. It is easy to verify that $M$ satisfy Assumption 3.1 and Assumption 3.2.

Since the adversarial reward functions are fixed, we know that the optimal hind-sight policy of $M$ is the optimal policy of $M'$. Thus, the adversarial MDP will degenerate to a non-adversarial MDP. The adversarial regret of algorithm on $M$ will also be identical to the non-adversarial regret on $M'$. By Theorem 5.4 in Zhou & Gu 2022, we know that when $K > \max\{3d^2, (d-1)/(192(b-1))\}$, for any algorithm, there exists a $B$-bounded homogeneous linear mixture MDPs with adversarial rewards such that the expected regret $\mathbb{E}[\text{Regret}(K)]$ is lower bounded by $d\sqrt{K}/(16\sqrt{3})$. □

## D Auxiliary Lemmas

**Lemma D.1** (Azuma-Hoeffding inequality, Azuma 1967). Let $M > 0$ be a constant. Let $\{x_i\}_{i=1}^n$ be a stochastic process, $\mathcal{G}_i = \sigma(x_1, \ldots, x_i)$ be the $\sigma$-algebra of $x_1, \ldots, x_i$. Suppose $\mathbb{E}[x_i|\mathcal{G}_{i-1}] = 0$, $|x_i| \leq M$ almost surely. Then, for any $0 < \delta < 1$, we have

$$\mathbb{P}\left(\sum_{i=1}^n x_i \leq M\sqrt{2n \log(1/\delta)}\right) > 1 - \delta.$$

**Lemma D.2** (Lemma 12, Abbasi-Yadkori et al. 2011). Suppose $\mathbf{A}, \mathbf{B} \in \mathbb{R}^{d \times d}$ are two positive definite matrices satisfying $\mathbf{A} \succeq \mathbf{B}$, then for any $\mathbf{x} \in \mathbb{R}^d$, $\|\mathbf{x}\|_{\mathbf{A}} \leq \|\mathbf{x}\|_{\mathbf{B}} \cdot \sqrt{\det(\mathbf{A})/\det(\mathbf{B})}$.

**Lemma D.3** (Lemma 11, Zhang et al. 2021b). Let $M > 0$ be a constant. Let $\{x_i\}_{i=1}^n$ be a stochastic process, $\mathcal{G}_i = \sigma(x_1, \ldots, x_i)$ be the $\sigma$-algebra of $x_1, \ldots, x_i$. Suppose $\mathbb{E}[x_i|\mathcal{G}_{i-1}] = 0$, $|x_i| \leq M$ and $\mathbb{E}[x_i^2|\mathcal{G}_{i-1}] < \infty$ almost surely. Then, for any $\delta, \epsilon > 0$, we have

$$\mathbb{P}\left(\left|\sum_{i=1}^n x_i\right| \leq 2\sqrt{2\log(1/\delta)\sum_{i=1}^n \mathbb{E}[x_i^2|\mathcal{G}_{i-1}]} + 2\sqrt{\log(1/\delta)}\epsilon + 2M\log(1/\delta)\right)$$
$$> 1 - 2(\log(M^2 n/\epsilon^2) + 1)\delta.$$

**Lemma D.4** (Lemma 12, Zhang et al. 2021b). Let $\lambda_1, \lambda_2, \lambda_4 > 0$, $\lambda_3 \geq 1$ and $\kappa = \max\{\log_2 \lambda_1, 1\}$. Let $a_1, \ldots, a_\kappa$ be non-negative real numbers such that $a_i \leq \min\{\lambda_1, \lambda_2\sqrt{a_i + a_{i+1} + 2^{i+1}\lambda_3} + \lambda_4\}$ for any $1 \leq i \leq \kappa$. Let $a_{\kappa+1} = \lambda_1$. Then we have $a_1 \leq 22\lambda_2^2 + 6\lambda_4 + 4\lambda_2\sqrt{2\lambda_3}$.

# E   INTUITION BEHIND DEFINITION 4.1 AND COMPUTATIONAL ISSUES OF LINE 3 IN ALGORITHM 2

The second and the third constraints in Definition 4.1 follows (3.2) and (3.1), which implies that the total probability of every $z_h$, i.e., $\sum_{s,a,s'\in\mathcal{S}\times\mathcal{A}\times\mathcal{S}} z_h(s,a,s')$, is 1. The last constraint under linear function approximation basicly induces an imagined transition function from $\phi(\cdot|\cdot,\cdot)$. At first glance, it is not obvious whether a Bregman projection step onto $\mathcal{D}_k$ (Line 3 of Algorithm 2) is computationally efficient. Despite such a projection cannot be formulated as a linear program, we can show $\mathcal{D}_k$ to be an intersection of convex sets of explicit linear or quadratic forms, on which the *Bregman projection onto convex sets* problem can be implemented by Dysktra's algorithm efficiently. A detailed discussion is given as follows.

First we provide an closed-form expression of the only implicit constraint in Definition 4.1.

**Lemma E.1.** For every $(s, a, h) \in \mathcal{S} \times \mathcal{A} \times [H]$, let $\mathbf{z}_{h,s,a}$ denote the vector of occupancy measure $z_h(s, a, \cdot)$ and $\boldsymbol{B}_{s,a} \in \mathbb{R}^{|\mathcal{S}|\times d}$ denote the matrix generated by stacking $\phi(\cdot|s,a)^\top$, i.e.

$$\mathbf{z}_{h,s,a} = z_h(s,a,\cdot) := \begin{bmatrix} z_h(s,a,s_{(1)}) \\ \vdots \\ z_h(s,a,s_{(||\mathcal{S}|)}) \end{bmatrix}, \boldsymbol{B}_{s,a} := \begin{bmatrix} \phi(s_{(1)}|s,a)^\top \\ \vdots \\ \phi(s_{(|\mathcal{S}|)}|s,a)^\top \end{bmatrix}, \tag{E.1}$$

where $\{(1), \ldots, (|\mathcal{S}|)\}$ is a indices set[5] of all states, then the only constraint including explicitly $\boldsymbol{\theta}_{s,a,h,k}$ in Definition 4.1 is equivalent to the following closed-form:

$$\left\|(\boldsymbol{B}_{s,a}\boldsymbol{\Sigma}_{k,0}^{-1/2})^\dagger(\mathbf{z}_{h,s,a} - \|\mathbf{z}_{h,s,a}\|_1\boldsymbol{B}_{s,a}\widehat{\boldsymbol{\theta}}_{k,0})\right\|_2 \leq \|\mathbf{z}_{h,s,a}\|_1\widehat{\beta}_k, \forall(s,a,h) \in \mathcal{S}\times\mathcal{A}\times[H] \tag{E.2}$$

*Proof.* Given $(s, a, h) \in S \times A \times [H]$, if $\sum_{s'\in S} z_h(s, a, s') = 0$, then obviously it satisfy (E.2). Now we consider the case that $\sum_{s'\in S} z_h(s, a, s') > 0$, then we denote $\mathbf{p}$ to be the normalized vector, i.e. $p_h(s, a, r) = z_h(s, a, r)/\sum_{s'\in S} z_h(s, a, s')$. Then, our new constraint is equivalent to

$$\left\|(\boldsymbol{B}_{s,a}\boldsymbol{\Sigma}_{k,0}^{-1/2})^\dagger(\mathbf{p} - \boldsymbol{B}_{s,a}\widehat{\boldsymbol{\theta}}_{k,0})\right\|_2 \leq \widehat{\beta}_k \tag{E.3}$$

and our original constraint becomes:

$$\exists\,\bar{\boldsymbol{\theta}} \in \mathcal{C}_k, \text{s.t.}, \mathbf{p} = \boldsymbol{B}_{s,a}\bar{\boldsymbol{\theta}}$$

which is equivalent to

$$\exists\,\bar{\boldsymbol{\theta}} \in \mathcal{C}_k, \text{s.t.}, \mathbf{p} - \boldsymbol{B}_{s,a}\widehat{\boldsymbol{\theta}}_{k,0} = \boldsymbol{B}_{s,a}\boldsymbol{\Sigma}_{k,0}^{1/2}[\boldsymbol{\Sigma}_{k,0}^{-1/2}(\bar{\boldsymbol{\theta}} - \widehat{\boldsymbol{\theta}}_{k,0})].$$

By definition of our confidence set, we know that $\bar{\boldsymbol{\theta}} \in \mathcal{C}_k$ means $\left\|\boldsymbol{\Sigma}_{k,0}^{-1/2}(\bar{\boldsymbol{\theta}} - \widehat{\boldsymbol{\theta}}_{k,0})\right\|_2 \leq \widehat{\beta}_k$, so this is the same as that the following function has a solution with norm less than $\widehat{\beta}_k$. In other word, this means that the solution with the least norm has a norm no bigger than $\widehat{\beta}_k$:

$$\mathbf{p} - \boldsymbol{B}_{s,a}\widehat{\boldsymbol{\theta}}_{k,0} = \boldsymbol{B}_{s,a}\boldsymbol{\Sigma}_{k,0}^{1/2}\mathbf{x}, \tag{E.4}$$

---

[5]In this paper, $s_i$ means the $i$-th state visited in an episode, while $s_{(i)}, i = 1, \ldots, |\mathcal{S}|$ is irrelevant to the episodic learning setting and only denotes the indexing order when we refer to the wildcard $\cdot \in \mathcal{S}$ in a vectorized notation.

---

**Algorithm 4** Dykstra algorithm with Bregman projections

---

**Require:** $\epsilon > 0$, $\Phi$, as defined in (4.2), which is strictly convex; $N$ closed convex sets $C_1, \ldots, C_N$, corresponding to the decomposition in (E.5), $C := \cap_i C_i \neq \emptyset$; $x_0 \leftarrow w^k$, where $w^k$ is defined in line 3 of Algorithm 2; $q_{-(N-1)} := \ldots := q_{-1} := q_0 := \mathbf{0} \in \mathbb{R}^{|\mathcal{S}|^2 |\mathcal{A}| H}$ serves as an auxiliary initialization.

1: **repeat**
2: $\quad x_n \leftarrow (P_n \circ \nabla \Phi^*)(\nabla f(x_{n-1}) + q_{n-N})$;
3: $\quad q_n \leftarrow \nabla f(x_{n-1}) + q_{n-N} - \nabla f(x_n)$;
4: **until** $||x_n - x_{n-1}||_{\text{TV}} \leq \epsilon$

---

where $\mathbf{x}$ is the unknown variable. The least norm solution of (E.4) is $(\boldsymbol{B}_{s,a} \boldsymbol{\Sigma}_{k,0}^{-1/2})^\dagger (\mathbf{p} - \boldsymbol{B}_{s,a} \widehat{\boldsymbol{\theta}}_{k,0})$, which should have a norm no bigger than $\widehat{\beta}_k$, and thus yields (E.3). Therefore, we conclude that the two constraints are equivalent.

$\square$

By Definition 4.1 and Lemma E.1, $\mathcal{D}_k$ can essentially be reformulated as the joint of several "easier" closed convex sets:

$$
\mathcal{D}_k = \Big\{ z_h(\cdot, \cdot, \cdot) \in \mathbb{R}^{|\mathcal{S}|^2 |\mathcal{A}|}, h \in [H] \mid
$$
$$
\sum_{s,a} z_h(s, a, s') = \sum_{a, s''} z_h(s', a, s''), \forall h \in [2:H] \Big\} \bigcap \Big\{
$$
$$
z_h(\cdot, \cdot, \cdot) \in \mathbb{R}^{|\mathcal{S}|^2 |\mathcal{A}|}, h \in [H] \mid
$$
$$
\sum_{a, s'} z_1(s, a, s') = \mathbb{1}\{s = s_1\} \Big\} \bigcap \Big\{
$$
$$
z_h(\cdot, \cdot, \cdot) \in \mathbb{R}^{|\mathcal{S}|^2 |\mathcal{A}|}, h \in [H] \mid
$$
$$
z_h(\cdot, \cdot, \cdot) \geq 0 \Big\} \bigcap \Big(
$$
$$
\bigcap_{(s,a,h) \in \mathcal{S} \times \mathcal{A} \times [H]} \Big\{ z_{h'}(\cdot, \cdot, \cdot) \in \mathbb{R}^{|\mathcal{S}|^2 |\mathcal{A}|}, h' \in [H] \mid
$$
$$
\big\|(\boldsymbol{B}_{s,a} \boldsymbol{\Sigma}_k^{-1/2})^\dagger (\mathbf{z}_{h,s,a} - \|\mathbf{z}_{h,s,a}\|_1 \boldsymbol{B}_{s,a} \widehat{\boldsymbol{\theta}}_{k,0}) \big\|_2 \leq \|\mathbf{z}_{h,s,a}\|_1 \widehat{\beta}_k, \Big\} \Big).
$$
(E.5)

Therefore, the *best approximation problem w.r.t Bregman divergence*[6] step, i.e. line 3 in Algorithm 2 can be cast to the *projection onto convex sets under Bregman divergence* (POCS (Bauschke & Borwein, 1996)) problem. Since $\mathcal{D}_k$ is the intersection of several **hyperplanes**, **halfspaces**, and **ellipsoids**[7], onto which (Bregman) projections are hopefully easier to conduct, the *Dykstra algorithm with Bregman projections* (Censor & Reich, 1998), which is verified to be convergent for general closed convex constraints (Bauschke & Lewis, 2000), can be utilized.

For the implementation of line 2 in Algorithm 4, a specialized scheme employing the *Dykstra algorithm with Bregman projections* may invoke the *projected gradient descent* algorithm to deal with the information projection subproblems onto hyperplanes and halfspaces, both of which are blessed with closed-form Euclidean projection formulas (see Lemma E.3); and invoke *Frank-Wolfe* to address the information projection subproblems onto ellipsoids, which only requires an efficient implementation of a linear optimization problem over the quadratic constraint, in that linear optimization over an ellipsoid has a closed-form formula (see Lemma E.4).

**Remark E.2.** The number of variables in line 3 of Algorithm 2 is of order $O(|\mathcal{S}|^2 |\mathcal{A}|)$, while its dual problem can not be much easier. The inequality constraints in (E.5) must be conducted for each $(s, a, h)$, i.e. the unknown transition kernel incurs at least $|\mathcal{S}||\mathcal{A}|H$ dual variables in the dual problem.

---

[6]In our case, it is just the information projection(Cover, 1999)

[7]Rigorously speaking, (E.2) can be relaxed to an elliptical constraint, because we only concern about $\mathbf{z}_{h,s,a}$ with $\|\mathbf{z}_{h,s,a}\|_1 \neq 0$. For $(h, s, a)$ whose $\|\mathbf{z}_{h,s,a}\|_1 = 0$, its induced transition kernel $\mathbb{P}_h(\cdot|s, a)$ can be any eligible unit simplex, which doesn't need to follow (3.4) in Lemma 3.3.

**Lemma E.3.** If $\mathbf{A} \in \mathbb{R}^{m \times n}$ is of full row rank, $\mathbf{b} \in \mathbb{R}^m$, $\mathbf{c} \in \mathbb{R}^n \backslash \{\mathbf{0}\}$, $d \in \mathbb{R}$, the orthogonal (Euclidean) projections of $\mathbf{x} \in \mathbb{R}^n$ onto $\{\mathbf{x} : \mathbf{A}\mathbf{x} = \mathbf{b}\}$ and $\{\mathbf{x} : \mathbf{c}^\top \mathbf{x} \le d\}$ are unique respectively, and have closed-form solutions as follows:

$$\mathbf{x} - \mathbf{A}^\top (\mathbf{A}\mathbf{A}^\top)^{-1} (\mathbf{A}\mathbf{x} - \mathbf{b}) = \underset{\mathbf{y}:\mathbf{A}\mathbf{y}=\mathbf{b}}{\mathrm{argmin}} \, \|\mathbf{y} - \mathbf{x}\|_2$$

$$\mathbf{x} - \frac{[\mathbf{c}^\top \mathbf{x} - d]_+}{\|\mathbf{c}\|_2^2} c = \underset{\mathbf{y}:\mathbf{c}^\top \mathbf{y}=d}{\mathrm{argmin}} \, \|\mathbf{y} - \mathbf{x}\|_2$$

**Lemma E.4.** If $\mathbf{A} \succ \mathbf{0}$, then linear optimization over an ellipsoid defined by $A \in \mathcal{S}_{++}^n$ and $x \in \mathbb{R}^n$:

$$\max_{\mathbf{y}} \mathbf{c}^\top \mathbf{y}$$

$$\text{s.t. } \|\mathbf{y} - \mathbf{x}\|_{\mathbf{A}^{-1}} \le 1,$$

has the unique solution with closed-form expression:

$$y = \mathbf{x} + \frac{\mathbf{A}\mathbf{c}}{\sqrt{\mathbf{c}^\top \mathbf{A}\mathbf{c}}}.$$

