# OpenReview forum: "Horizon-free Reinforcement Learning in Adversarial Linear Mixture MDPs"
_ICLR.cc/2024/Conference — ICLR 2024 poster_

### Official Review · Reviewer_K64K · 2023-10-19

**Soundness:** 3 good
**Presentation:** 3 good
**Contribution:** 3 good
**Rating:** 6
**Confidence:** 4

**Summary:**

This paper studies horizon-free RL in adversarial linear mixture MDPs with full-information feedback. This paper proposes an algorithm that employs a variance-aware weighted least square for the transition kernel and an occupancy measure-based method for the online search of a stochastic policy. The authors show the algorithm achieves a regret with polylogarithmic dependence on $H$. Further, this paper provides a lower bound showing the inevitable polylogarithmic dependence on state number $S$.

**Strengths:**

1. The paper is the first work that studies near-optimal horizon-free RL algorithms under adversarial reward and linear function approximation. This progress deserves to be known to the community.
2. The connection between the value function derived from occupancy measure guided policy updating and the other one derived from backward iteration (Lemma 6.1) is new as far as I know, which may inspire other studies for RL problems.
3. The paper is clearly written and well-organized. The proofs are technical sound though I don't check the proofs.

**Weaknesses:**

1. The novelty of this paper may be limited. Most of the analysis follows from that of horizon-free reinforcement learning for linear mixture MDPs with stochastic rewards (Zhou and Gu, 2022).
2. The occupancy measure-based algorithm is not computationally efficient as the running time has polynomial dependence on the state number $S$ and action number $A$.

**Questions:**

N/A

---

> ### Author Response · Authors · 2023-11-19
>
> Thank you for your supportive feedback. We will answer your question as follows:
>
> **Q1**: The novelty of this paper may be limited. Most of the analysis follows from that of horizon-free reinforcement learning for linear mixture MDPs with stochastic rewards [1].
>
> **A1**: We agree that the our analysis framework follows [1]. However, compared to [1] and other very related work [2][3], our paper actually introduces several new techniques. First, compared to [2][3], we use occupancy-measure, rather than policy-optimization, to update the policy, making our work the first to apply occupancy-measure to this setting. Second, compared to [1], we adopted random policy, which inevitably introduce a new policy noise term $\sum_{k=1}^K\sum_{h=1}^H \mathbb{E}\big[Q_{k, h}(s_h^k, a) - Q_{k, h}^{\pi^k}(s_h^k, a)\big] - \big(Q_{k,h}(s_h^k, a_h^k) - Q_{k, h}^{\pi^k}(s_h^k, a_h^k) \big)$ with $a\sim\pi^k_h(\cdot|s_h^k)$. This term is bounded by $\tilde{O}(\sqrt{KH})$ by using Azuma-Hoeffding's inequality and therefore introduces an unwanted dependency on $H$. To tackle this issue, we introduce a notion of conditional variance of $Q_{k, h+1}^{2^m}(s_{h+1}^k, a)$ into the analysis of Algorithm 3 (HOME) to bound the policy noise in a recursive way and settle the poly$(H)$ dependency down. This is also a new technique that does not appear in previous analysis [1]. Third, we also proposed a novel lower bound showing that horizon-free regret can only be achieved under finite state assumption.
>
> **Q2**: The occupancy measure-based algorithm is not computationally efficient as the running time has polynomial dependence on the state number S and action number A.
>
> **A2**: We agree with the reviewer as we have discussed in Remark E.2. The necessity of computing all the $\Omega(S)$ entries of occupancy measure is the limitation of our algorithm. However, as we have shown in Section 6, using occupancy measure is the key technique to get rid of the $H$ dependency in online mirror descent regret. We will explore more efficient algorithms in the future work.
>
> References:
>
> [1] Zhou, D., & Gu, Q. (2022). Computationally efficient horizon-free reinforcement learning for linear mixture mdps. arXiv preprint arXiv:2205.11507.
>
> [2] He, J., Zhou, D., & Gu, Q. (2022, May). Near-optimal policy optimization algorithms for learning adversarial linear mixture mdps. In International Conference on Artificial Intelligence and Statistics (pp. 4259-4280). PMLR.
>
> [3] Cai, Q., Yang, Z., Jin, C., & Wang, Z. (2020, November). Provably efficient exploration in policy optimization. In International Conference on Machine Learning (pp. 1283-1294). PMLR.

---

### Official Review · Reviewer_ZMLZ · 2023-10-25

**Soundness:** 3 good
**Presentation:** 3 good
**Contribution:** 3 good
**Rating:** 6
**Confidence:** 3

**Summary:**

This paper introduces the first algorithm enjoying horizon free bounds for the adversarial linear mixture MDP. The algorithm is based on a careful combination of policy updates step performed with mirror descent steps in the occupancy measures space and an optimistic policy evaluation phase carried out using weighted ridge regression estimators.

An interesting finding is also a separation between the adversarial and non adversarial case. Indeed, the authors managed to prove an asymptothic lower bounds which shows that either $\sqrt{H}$ or $\log S$ must be paid in the regret bound while a $S$ independent horizon free regret upper bound can be obtained in the non adversarial case.

**Strengths:**

I think that the algorithm is original and well explained in the main text.

The result which entails a $\log S$ dependence would not be very satisfactory in the function approximation setting but the author nicely shows that either this dependence or a $\sqrt{H}$ dependence must be suffered.

I also enjoyed the lower bound construction which considers a tabular deterministic MDP and reduces it to an expert problem.

The proofs look correct to me.

**Weaknesses:**

There are few clarity omission or missing definition in the submission. Hereafter, I list few of them:

- I think it should be clearer that also homogenous transition dynamics are required for obtaining reward free bounds. Therefore, the Bellman optimality equation at page 3 should not have $h$ in the footnote of the operator $\mathbb{P}$.

- the value function $\overline{V_{k,1}}$ is never formally defined in the paper. So it is difficult to understand what it denotes when reading the regret decomposition in equation (6.1).
If I understood correctly from the Appendix, each mirror descent iterate $z_k$ induces via the marginals a transition kernel $\overline{p_{k}}$ and a policy $\pi_k$. At this point $\overline{V_{k,1}}$ denotes the initial value of policy $\pi_k$ in the MDP endowed with reward function $r_k$ and transition dynamics $\bar{p}_k$. Can the authors confirm that this is correct and if yes add it to their revision ?

- The definition of Regret at page 3 is a bit unclear. Indeed saying that $V^\star_k$ is the optimal state value function could make the reader believe that $V^\star_k = \max_{\pi} V^{\pi}_k$, that is the regret we control has the maximum inside the sum.
However, the regret controlled in the paper has a fixed comparator policy which might not be optimal for any of the reward function revealed at each round.

**Questions:**

I think that it is unclear that $I^k_h$ defined in Appendix C.2 is decreasing. After inspecting the proofs I think that what the authors need is that for any fixed $k$ than $I^k_h$ is decreasing with respect to $h$. Is this correct ?

---

> ### Author Response · Authors · 2023-11-19
>
> Thank you for your positive and constructive feedback. We answer your questions as follows:
>
> **Q1**:I think it should be clearer that also homogenous transition dynamics are required for obtaining reward free bounds. Therefore, the Bellman optimality equation at page 3 should not have ℎ in the footnote of the operator P.
>
> **A1**: Thank you for your suggestion and we have fixed it in the reversion.
>
> **Q2**: The value function $\bar{V}_k,1$ is never formally defined in the paper.
>
> **A2**: We are sorry for the missing formal definition. We have added an equation to formally define $\bar{V}_k,1$ in the revision.
>
> **Q3**: The definition of Regret at page 3 is a bit unclear.
>
> **A3**: Thank you for your suggestion and we have revised it to make it clearer.
>
> **Q4**: I think that it is unclear that $I_k^h$ defined in Appendix C.2 is decreasing. After inspecting the proofs I think that what the authors need is that for any fixed k  $I_k^h$ is decreasing with respect to $h$. Is this correct ?
>
> **A4**: Yes, we indeed need for any fixed k then $I_k^h$ is decreasing with respect to $h$. We apologize for the confusion and have made it clear in the revision.

---

> > ### Comment · Reviewer_ZMLZ · 2023-11-19
> >
> > Dear Authors,
> >
> > Thanks for correcting the issues I mentioned in my review.
> >
> > I am keeping my positive score and will support the acceptance of your manuscript in the discussion phase.
> >
> > Have a nice weekend,
> >
> > Best,
> > Reviewer

---

> > > ### Author Response · Authors · 2023-11-19
> > > **Thank you!**
> > >
> > > We're glad that our response has effectively addressed your concerns and suggestions.

---

### Official Review · Reviewer_tM5p · 2023-10-31

**Soundness:** 3 good
**Presentation:** 3 good
**Contribution:** 3 good
**Rating:** 6
**Confidence:** 3

**Summary:**

This paper addresses the question of whether the favorable polylogarithmic regret seen in reinforcement learning (RL) with respect to the planning horizon can also be extended to adversarial RL scenarios. The authors introduce the first horizon-independent policy search algorithm, designed to cope with challenges arising from exploration and adversarial reward selection over episodes. The algorithm utilizes a variance-uncertainty-aware weighted least square estimator for the transition kernel and an occupancy measure-based approach for online stochastic policy search.

**Strengths:**

Given my limited expertise in the adversarial RL domain, my evaluation focuses exclusively on the technical soundness and clarity of the paper. The manuscript exhibits a commendable standard of articulation. The framing of the problem, underlying assumptions, and derived outcomes are adequately elucidated. Notably, the inclusion of a proof sketch in Section 6 enhances the paper's comprehensibility, serving as a valuable reference point for those seeking deeper insight into the paper's theoretical foundations.

**Weaknesses:**

The paper makes relatively strong assumptions: linear, finite-state MDPs and full-information feedback. The only novel aspect here is the paper tackles adversarial reward functions rather than fixed or stochastic rewards. But even so, I think that the full-information feedback assumptions greatly alleviate the difficulty of adversarial rewards. To me, the hardness result is more interesting: an unbounded state space will incur regret in $\Omega(\sqrt{H})$. Is this result novel in the literature?

I am a bit confused about the assumptions about the reward. Firstly, can the rewards be negative? If so, would assumption 3.1 still make sense? Furthermore, is assumption 3.1 equivalent to the bounded rewards assumption, i.e., if rewards are bounded in $[-R, R]$, we can always scale everything by $1/RH$ to satisfy assumption 3.1.

**Questions:**

Please see the questions in the weaknesses section above. I am happy to increase my score if there are any misunderstandings.

---

> ### Author Response · Authors · 2023-11-19
>
> Thank you for your supportive feedback. We will answer your questions as follows.
>
> **Q1**: The paper makes relatively strong assumptions: linear, finite-state MDPs and full-information feedback. The only novel aspect here is the paper tackles adversarial reward functions rather than fixed or stochastic rewards.
>
> **A1**: We agree with the reviewer that the assumptions we made (linear, full-information) are relatively strong. However, even under these relatively strong but standard assumptions [1][2][3], how to achieve horizon-free regret is still an open problem before our work. Notably, our result is the first horizon-free algorithm in this setting. We agree that finite state space is an extra assumption compared to previous works. However, this is a necessary assumption to achieve a horizon-free regret with adversarial reward, as we have shown in the hardness result in Theorem 5.3.
>
> **Q2**: To me, the hardness result is more interesting: an unbounded state space will incur regret in $\Omega(\sqrt{H})$. Is this result novel in the literature?
>
> **A2**: Thank you for your interest! To the best of our knowledge, this result is new and we are the first to show that an infinitely large state space is incompatible with horizon-free regret under adversarial rewards. Therefore, we believe this result is novel and a noteworthy contribution.
>
> **Q3**: Can the rewards be negative?
>
> **A3**: Yes, technically, the reward can be negtive. To accommodate this, we can scale up the reward by a factor of 2 and subsequently translate it to the interval $[-1/H, 1/H]$ by subtracting a constant of $1/H$.
>
> **Q4**: Furthermore, is assumption 3.1 equivalent to the bounded rewards assumption, i.e., if rewards are bounded in [−R,R], we can always scale everything by 1/RH to satisfy assumption 3.1.
>
> **A4**: Yes, scaling the bounded rewards in [-R, R] by a constant of $1/RH$ can make it satisfy our assumption. We would like to bring to the reviewer's attention that the dependency on $H$ in the regret bound of RL algorithms has two sources: 1) the length of planning horizon, and 2) scale of the rewards. Here to offset the dependency on $H$ contributed by the scale of the reward, we scale down the reward by a factor $H$ so that the total reward in each episode is bounded by one, as suggested by [4].
>
> References:
>
> [1] Zhou, D., & Gu, Q. (2022). Computationally efficient horizon-free reinforcement learning for linear mixture mdps. arXiv preprint arXiv:2205.11507.
>
> [2] Cai, Q., Yang, Z., Jin, C., & Wang, Z. (2020, November). Provably efficient exploration in policy optimization. In International Conference on Machine Learning (pp. 1283-1294). PMLR.
>
> [3] He, J., Zhou, D., & Gu, Q. (2022, May). Near-optimal policy optimization algorithms for learning adversarial linear mixture mdps. In International Conference on Artificial Intelligence and Statistics (pp. 4259-4280). PMLR.
>
> [4] Nan Jiang and Alekh Agarwal. Open problem: The dependence of sample complexity lower bounds on planning horizon. In Conference On Learning Theory, pp. 3395–3398. PMLR, 2018.

---

> > ### Comment · Reviewer_tM5p · 2023-11-22
> > **Thank you**
> >
> > Dear authors,
> >
> > Thank you for your response. I would like to keep my original score and recommend acceptance of the paper.
> >
> > Warmest regards,
> > Reviewer.

---

> > > ### Author Response · Authors · 2023-11-22
> > > **Thank you**
> > >
> > > Thank you for your support! We're glad that our response has addressed all your questions.

---

### Official Review · Reviewer_p5dn · 2023-11-01

**Soundness:** 3 good
**Presentation:** 3 good
**Contribution:** 2 fair
**Rating:** 6
**Confidence:** 4

**Summary:**

This paper studies the online learning problem of horizon-free and linear mixture Markov Decision Processes (MDPs).
To the best of my knowledge, this is the first paper that can achieve theoretical guarantees with adversarial losses, that is, the loss function can change arbitrarily from episode to episode.
To achieve this result, the authors propose two main techniques: (1) a variance-uncertainty-aware weighted least square estimator and (2) an occupancy measure-based approach for constructing policies. The first technique is widely use for linear mixture MDPs, while the second one is mainly used for adversarial losses.
Combining these two techniques to establish valid regret guarantees is quite challenging.
More importantly, the final regret bound is of the order $O(d\sqrt{K})$, which is nearly the optimal.

**Strengths:**

1. The idea of combining the two techniques is very interesting. It would be great to see such combination to be applied in other (more general) linear MDP settings.
2. Though I just skimmed the proof of several lemmas, the results seems to be rigorous proved and mathematically correct.

**Weaknesses:**

This paper does not have any specific weaknesses.

**Questions:**

1.Is it possible to design policy optimization algorithms for this problem setting?
2.Is it possible to avoid the usage of occupancy measure (which is not quite efficient in real world).

---

> ### Author Response · Authors · 2023-11-19
>
> Thank you for your positive feedback. We answer your questions as follows:
>
> **Q1**: Is it possible to design policy optimization algorithms for this problem setting?
>
> **A1**: We would like to clarify that policy optimization has previously been applied to adversarial linear mixture MDP [1][2]. However, none of these approaches can guarantee a horizon-free regret upper bound. The reason for this limitation has been shown in Section 6. Specifically, the regret for online mirror descent is $\tilde{O}(H\bar{L}\sqrt{K})$, where $\bar{L}$ is the upper bound of the subgradient. When employing (multiplicative weights update/EXP3 type) policy optimization, due to reward accumulating, the scale of the Q-functions and the correspoding subgradient is $\bar{L}=O(1)$, which will inevitably introduce a dependency on $H$. Therefore, we resort to occupancy measures to tackle this challenge.
>
> **Q2**: Is it possible to avoid the usage of occupancy measure (which is not quite efficient in real world).
>
> **A2**: To our knowledge, to tackle adversarial reward, existing approaches are based on either policy optimization or occupancy-measure. As we found that policy optimization is difficult to deduce a horizon-free guarantee, we show that a feasible way is to take occupancy measure-based approach. We will explore other approaches rather than occupancy measure in the future work.
>
> References:
>
> [1] Cai, Q., Yang, Z., Jin, C., & Wang, Z. (2020, November). Provably efficient exploration in policy optimization. In International Conference on Machine Learning (pp. 1283-1294). PMLR.
>
> [2] He, J., Zhou, D., & Gu, Q. (2022, May). Near-optimal policy optimization algorithms for learning adversarial linear mixture mdps. In International Conference on Artificial Intelligence and Statistics (pp. 4259-4280). PMLR.

---

### Meta-Review · Area_Chair_x8KJ · 2023-12-03

**Metareview:**

This paper studies low-regret algorithm for Linear Mixture MDPs under adversarial losses. It proposed the first algorithm that achieves horizon-free regret on the order of $\tilde O(d\sqrt{K})$, which is near optimal and a solid theoretical contribution. This paper also presents a lower-bound construction, showing that an infinitely large state space is incompatible with horizon-free regret under adversarial rewards, an interesting trade-off not previously known for the stochastic reward setting.

Overall, this is a very solid theoretical RL paper.

**Justification For Why Not Higher Score:**

Given the strong theoretical focus and the very specific setting that is being studied here, it might not be suitable for oral/spotlight presentation.

**Justification For Why Not Lower Score:**

NA

---

### Decision · Program_Chairs · 2024-01-16

Accept (poster)